# UniE2F: A Unified Framework for Event-to-Frame Reconstruction with Diffusion Model

## Abstract

Event cameras excel at high-speed, low-power, and high-dynamic-range scene perception. However, as they fundamentally record only relative intensity changes rather than absolute intensity, the resulting data streams suffer from a significant loss of spatial information and static texture details. In this paper, we address this limitation by leveraging the generative prior of a pre-trained video diffusion model to reconstruct high-fidelity video frames from sparse event data. Specifically, we first establish a baseline model by directly applying event data as a condition to synthesize videos. Then, based on the physical correlation between the event stream and video frames, we further introduce the event-based inter-frame residual guidance to enhance the accuracy of video frame reconstruction. Furthermore, we extend our method to video frame interpolation and prediction in a zero-shot manner by modulating the reverse diffusion sampling process, thereby creating a unified event-to-frame reconstruction framework. Experimental results on real-world and synthetic datasets demonstrate that our method significantly outperforms previous approaches both quantitatively and qualitatively. The code will be publicly available.

## 1 Introduction

Event cameras, also known as dynamic vision sensors, measure high-frequency changes in pixel intensity as "events" and output a continuous flow encoding the time, location, and polarity of each change (Brandli et al., 2014a;b). This asynchronous processing allows them to capture high dynamic range scenes (up to 140 dB) with exceptional temporal resolution (approx. 1 μs) and low power consumption (5 mW) without motion blur. While these attributes make event cameras ideal for high frame rate video reconstruction (Rebecq et al., 2019; Gallego et al., 2020), the data is inherently sparse as it only captures relative brightness changes. Consequently, this limited information content has led previous approaches (Rebecq et al., 2019; Stoffregen et al., 2020; Scheerlinck et al., 2020; Cadena et al., 2021; Weng et al., 2021) to reconstruct images that differ significantly from the richly detailed scenes observed in real-world environments. Moreover, the current utility of event cameras extends beyond image reconstruction. Their microsecond-level resolution unlocks critical applications in temporal modeling, specifically Video Frame Interpolation (VFI) (Dong et al., 2023) and Video Frame Prediction (VFP) (Li et al., 2020). Where standard cameras struggle with high-speed motion often suffering from blur and temporal gaps, event-based VFI (Tulyakov et al., 2021; 2022a) bridges these discontinuities by leveraging continuous event streams to synthesize intermediate frames. This enables applications ranging from smooth slow-motion smartphone photography to latency-critical tasks in autonomous navigation. Similarly, VFP (Zhu et al., 2024a; Wang et al., 2025) utilizes event dynamics to forecast future states, compensating for hardware limitations in scientific observation. However, due to the limited model capability, previous work usually process those tasks of reconstruction, VFI, and VFP have been treated as isolated tasks.

Recently, the diffusion model (Ho et al., 2020) has shown remarkable progress in image generation and image restoration (Kawar et al., 2022; Chung et al., 2023; Wang et al., 2023; Saharia et al., 2022b;a; Gao et al., 2023), relying on a core mechanism of iterative noise addition and removal to generate realistic outcomes. Through large-scale pre-training, the stable diffusion (Rombach et al., 2022) model has accumulated rich generative prior knowledge and a powerful capability to approximate diverse and complicated distributions. With the aid of conditional information, such as text or image prompts, the stable diffusion model (Rombach et al., 2022) can achieve a more

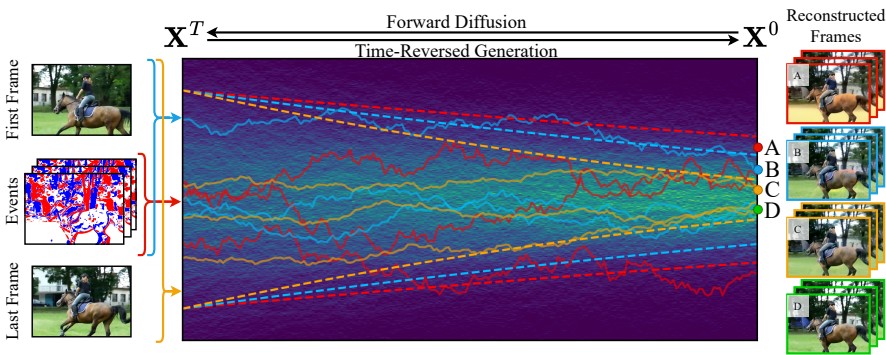

Figure 1: Illustration of the forward and backward diffusion processes for our UniE2F under the conditional event data. The right and left parts indicate the inputs and results of our algorithm, while in the central plot, the solid and dashed lines with the same color represent the reverse-time sampling SDE and ODE trajectories under the same setting, respectively. The proposed method can adapt to different types of event-assisted frame reconstruction tasks. (**A**) **event-based frame reconstruction**: with input of only event, to reconstruct RGB frame; (**B**) **frame prediction**: with input of both event and the first frame to reconstruct the remaining frames; and (**C**) **frame interpolation**: with input of event and the first and last frames to reconstruct the intermediate frames. (**D**) denotes the ground-truth frames corresponding to the conditional event data.

accurate generation process, thereby enhancing the fidelity of details and semantic consistency of the generated content. Furthermore, the stable video diffusion (SVD) model (Blattmann et al., 2023a), benefiting from large-scale pre-training on video data, also demonstrates its capabilities to produce videos that are both realistic and visually pleasing based on text and image conditions.

In this paper, building on the powerful generative capacity of the SVD model, we propose the **Uni**fied Framework for **E**vent**2F**rame (UniE2F), which leverages event data as the conditional input to guide the reconstruction process, bridging the gap between sparse event data and highly detailed real-world scenarios. After fine-tuning the SVD model, we introduce the event-based inter-frame residual guidance, which exploits event data to effectively constrain the residual between consecutive reconstructed frames. Specifically, during the reverse diffusion process, this mechanism first predicts the inter-frame residuals from event data. Then, it iteratively refines the intermediate latent via the gradient descent algorithm based on the inter-frame residuals to improve the reconstruction accuracy. In addition to reconstructing video frames purely from the event data, we further adopt the proposed method to event-based video frame interpolation (Tulyakov et al., 2021; 2022b; He et al., 2022; Wu et al., 2022) and prediction (Zhu et al., 2024a) in a zero-shot manner.

Based on the powerful generative capacity of mapping sparse and asynchronous event data to continuous video frames, the prior information from the first and last frames for interpolation, or solely from the first frame for prediction, is utilized to guide the reverse diffusion process. Specifically, we modulate the score function by incorporating deviation derived from the discrepancies between the estimated clean latent and the given reference latent, guiding the reverse sampling process to reconstruct intermediate or subsequent frames with enhanced temporal consistency and visual fidelity. As shown in Fig. 1, as more prior information is provided, our UniE2F can produce frames that more closely match the ground truth. Through these designs, we construct a unified event-to-frame reconstruction framework that effectively handles diverse applications while reducing the need for task-specific models. Experimental results on real-world and synthetic datasets demonstrate that our method significantly outperforms previous approaches in both qualitative and quantitative assessments.

In summary, the main contributions of this work are as follows.

- We propose a diffusion-based event-to-frame reconstruction framework that utilizes the event-based inter-frame residual guidance to align the physical correlations between those frames, thus boosting the performance.

- We give theoretical analysis that optimizing the proposed regularization based event physical mechanism can indeed help minimize error upper boundary.

- We extend the proposed method to video frame interpolation and prediction in a *zero-shot* manner, which formulates the reverse sampling of the diffusion model to build a unified event-to-frame reconstruction framework.

## 2 RELATED WORK

**Event-based Video Frame Reconstruction.** Event cameras offer unique advantages, including high dynamic range, high temporal resolution, and low power consumption (Brandli et al., 2014a;b; Gallego et al., 2020). Existing event-based video frame reconstruction approaches can be broadly divided into two categories: diffusion-model-based methods and non-diffusion-based methods. Early non-diffusion methods relied on the intensity gradients provided by events and optical flow to reconstruct scene intensity (Cook et al., 2011; Kim et al., 2008; Munda et al., 2018; Bardow et al., 2016). Recently, deep learning has driven major advances: Rebecq et al. (2019) pioneered the use of convolutional recurrent neural networks for high-quality reconstructions; FireNet (Scheerlinck et al., 2020) achieved faster inference with a lightweight design; Stoffregen et al. (2020) improved model generalization via data augmentation; By utilizing the self-supervised framework (Paredes-Vallés & De Croon, 2021), spatially-adaptive denormalization (Cadena et al., 2021), transformer models (Weng et al., 2021), and context-guided hypernetworks, these approaches have shown remarkable performance. For the diffusion-based methods, based on the coarse reconstruction result from ETNet (Weng et al., 2021), Liang et al. (2023) and Zhu et al. (2024c) utilized a diffusion model (Ho et al., 2020) to enhance the high-frequency components of objects in reconstructed images. In addition, Chen et al. (2024) leveraged language guidance and pretrained diffusion models to achieve semantic-aware event-to-video reconstruction. In E2VIDiff, Liang et al. (2024) introduced diffusion models with event-guided sampling to reconstruct colorful and perceptually realistic videos from achromatic event streams. In CUBE, Zhao et al. (2024) utilized the edge information from events and combined it with textual descriptions to guide the diffusion network to synthesize videos in a zero-shot manner.

In contrast to these methods, which are limited to reconstructing video frames from events, our UniE2F, leveraging rich pre-trained generative priors and score function modulation, not only reconstructs video frames in real-world scenes from event inputs, but also performs video frame interpolation and future-frame prediction in a zero-shot manner, without any additional training.

**Event-based Video Frame Interpolation.** Due to the high temporal resolution and low latency of event data, it has been utilized to interpolate intermediate frames to achieve accurate and temporally consistent video frame reconstruction. In Time Lens, Tulyakov et al. (2021) introduced a unified CNN framework that combines event-based motion estimation with both warping- and synthesis-based interpolation to robustly generate high-quality frames under non-linear motion, motion blur, and illumination changes. Then, in Time Lens++ (Tulyakov et al., 2022c), the motion spline estimator and multi-scale feature fusion module were proposed to achieve temporally consistent interpolation and reduce ghosting artifacts. Liu et al. (2025) utilized the continuous trajectory guided motion estimation module to track the continuous motion trajectory of each divided patch. In TimeLens-XL, Ma et al. (2024) decomposed large inter-frame displacements into multiple small-step motions and recursively estimates the optical flow at each step using events, enabling tracking of nonlinear motion. Additionally, CBMNet (Kim et al., 2023) was proposed to leverages cross-modal asymmetric bidirectional motion fields from events and images to interpolate video frames without motion approximations. Based on the pretrained diffusion model, Chen et al. (2025) trained a trainable copy on a real-world and event-based interpolation dataset in order to control the diffusion network to synthesize intermediate frames.

Distinct from previous work, we propose a new paradigm that simply modulates the reverse diffusion sampling process to transfer reconstruction priors to interpolation and prediction in a zero-shot manner. By fully leveraging event streams, this unified paradigm completes event-driven reconstruction, interpolation, and prediction without task-specific datasets or complex network designs.

**Diffusion Models.** Denoising Diffusion Probabilistic Model (DDPM) (Ho et al., 2020) has demonstrated remarkable success in image synthesis, with its capability evidenced by generating high-quality images from random noise. The Latent Diffusion Model (LDM), proposed by Rombach et al. (2022), performs forward and reverse diffusion processes on the latent space, significantly enhancing efficiency in high-resolution image synthesis. Following LDM, Rombach et al. (2022) introduced DiffIR (Xia et al., 2023), applying diffusion on a compact prior representation to achieve

more efficient and stable image restoration. To enable high-resolution video generation, Blattmann et al. (2023b) developed Video LDM by introducing a temporal dimension to the pre-trained image LDM (Rombach et al., 2022) and fine-tuning it on video data. Subsequently, the SVD model is accomplished by pretraining Video LDM (Blattmann et al., 2023b) on well-curated datasets and pipelines, resulting in significant performance improvements for high-quality video generation.

**Event Stacking Approaches.** To convert the asynchronous event stream into a tensor suitable for convolutional networks, a common practice is to partition it into temporal bins, accumulate events in each bin to form "event frames," and stack these along the channel dimension into an event volume. Wang et al. (2019) first systematically introduced time-based and event-count-based stacking strategies (SBT/SBE) for event-based HDR and high frame-rate video reconstruction, establishing the basic form of event stacking; Nam et al. (2022) proposed mixed-density stacks in stereo event-based depth estimation to balance short-term details and long-term context; Teng et al. (2022) further modeled bidirectional event summations as a learnable Neural Event Stack for image enhancement, alleviating the sensitivity of hand-crafted stacks to noise.

## 3 PRELIMINARY

**Event Representation.** Given an event stream $\{e_i\}_{i=0}^{N_E-1}$ over a duration $T$, each event is $e_i = (x_i, y_i, t_i, p_i)$, where $(x_i, y_i)$ are pixel coordinates, $t_i$ is the timestamp, and $p_i \in +1, -1$ is the polarity. To interface with the frame-based diffusion model, we group events between adjacent target frame timestamps in $\{s_f\}_{f=0}^{F-1}$ into event group $\mathcal{G}_f = \{e_i \mid \mathbf{s}_{f-1} \leq t_i < \mathbf{s}_f\}$ (with $s_{-1} = 0$). Each event group $\mathcal{G}_f$ is then converted to a 3-channel tensor $\mathbf{E}_f \in \mathbb{R}^{3 \times H \times W}$ encoding: (i) sum of all events, (ii) sum of positive events, and (iii) sum of negative events. More details of the event representation are provided in Appendix **??**.

**Diffusion Model.** As a type of generative model, the diffusion model (Ho et al., 2020) first utilizes a forward diffusion process to transform data from complex high-dimensional distributions to simpler ones (typically Gaussian), and then, applies the reverse diffusion process to reconstruct the original data distribution from the simplified one. According to (Song et al., 2021), the forward stochastic differential equation (SDE) process of the latent diffusion model (Rombach et al., 2022) can be formulated as

$$d\mathbf{x} = f(t)\mathbf{x}dt + g(t)d\mathbf{w}, \tag{1}$$

where $\mathbf{x} \in \mathbb{R}^n$ is the noised latent state, $\mathbf{w} \in \mathbb{R}^n$ represents a standard Wiener process, $t$ indicates the diffusion timestamp, $f(t)$ and $g(t)$ yield the drift and diffusion coefficients, which indicate the variation of data and noise components during the diffusion process. Therefore, the reverse process of the latent diffusion model (Rombach et al., 2022) can be formulated via the ordinary differential equation (ODE) solution (Wu et al., 2024):

$$d\mathbf{x} = \left[ f(t)\mathbf{x} - \frac{1}{2}g^2(\mathbf{x})\nabla_{\mathbf{x}} \log q_t(\mathbf{x}) \right] dt, \tag{2}$$

where $\nabla_{\mathbf{x}} \log q_t(\mathbf{x})$ is usually approximated through training a score model $\mathcal{S}_\theta(\mathbf{x}, t)$ with parameter $\theta$. Taking into account the special case of the variance exploding (VE) diffusion process (Song et al., 2021) of the SVD model, the reverse process can be simplified as

$$\mathbf{x}^{t-1} = \mathbf{x}^t - \frac{\mathbf{x}^t - \mathbf{u}^t}{\sigma_t}(\sigma_t - \sigma_{t-1}), \tag{3}$$

where $\mathbf{u}^t = \mathcal{X}_\theta(\mathbf{x}^t, t)$ represents the estimated clean latent from the noised latent $\mathbf{x}^t \sim \mathcal{N}(\mathbf{x}^0, \sigma_t^2 \mathbf{I})$ at step $t$ by the denoising U-Net $\mathcal{X}_\theta(\cdot)$, and $\sigma_t$ is the variance of the Gaussian noise in the diffusion process.

## 4 PROPOSED METHOD

Leveraging the strong generative prior of the pre-trained stable video diffusion (SVD) model, we first reconstruct video frames solely from the event representation $\mathbf{E}$. When the first and/or last frames (optionally) of the ground-truth sequence $\mathbf{V}$ become available, we seamlessly adapt this event-driven reconstruction to frame interpolation and prediction in a zero-shot fashion, yielding a unified framework for event-to-frame (UniE2F), as illustrated in Fig. 2. Technically, we first

Figure 2: The schematic of the proposed framework, which integrates event-based inter-frame residual guidance during the inference stage. At step $t$ ($t \leq \tau$), given event representations, we utilize a ResNet to predict the inter-frame residuals between consecutive frames. Then, these residuals are utilized to formulate the inter-frame residual loss $\mathcal{L}_{\text{residual}}$, which is optimized via a gradient descent algorithm to update the noisy latent.

introduce a baseline event-based video frame reconstruction model by fine-tuning the SVD model using event representations as conditional inputs, providing a foundation for the subsequent design (Sec. 4.1). Then, we present the inter-frame residual guidance (Sec. 4.2). Finally, we modulate the score function to enable the adaptation to video frame interpolation and prediction in a *zero-shot* manner (Sec. 4.3).

## 4.1 FINE-TUNING WITH EVENT REPRESENTATION

To leverage the high temporal resolution property of event data to guide the pre-trained diffusion model with large generative priors for video frame reconstruction, we propose an event-conditioned fine-tuning strategy. Specifically, we first encode the event representation $\mathbf{E}$ with a dedicated encoder into $\mathcal{E}(\mathbf{E})$ that serves as the conditioning input. Subsequently, following the training scheme in previous works (Song et al., 2021; Karras et al., 2022; Wu et al., 2024), at each diffusion step $t$ we randomly select a noisy latent $\mathbf{X}^t \in \mathbb{R}^{F \times C \times H \times W}$ — and, for notational simplicity, treat both the latent and the video frame tensors as having the same size. Conditioned on the event representation, the denoising U-Net predicts the noise component $\epsilon$ to estimate the clean latent $\mathbf{U}^t = \mathcal{X}_\theta\left(\mathbf{X}^t, \mathcal{E}(\mathbf{E}), t\right)$. Then the diffusion model is fine-tuned by minimizing the discrepancy between the estimated clean latent $\mathbf{U}^t \in \mathbb{R}^{F \times C \times H \times W}$ and the ground truth clean latent $\mathbf{X}^0 = \mathcal{E}(\mathbf{V})$, with the loss function $\mathcal{L}_{\text{fine-tuning}}$ formulated as

$$\mathcal{L}_{\text{fine-tuning}} = \mathbb{E}_t \left\{ \lambda(t) \mathbb{E}_{\mathbf{X}^0} \mathbb{E}_{\mathbf{X}^t | \mathbf{X}^0} \left[ \left\| \mathbf{X}^0 - \mathbf{U}^t \right\|_2^2 \right] \right\}, \ \lambda(t) = \frac{\sigma_t^2 + \sigma_{data}^2}{\left(\sigma_t + \sigma_{data}\right)^2}, \tag{4}$$

where $\lambda(t)$ serves as a weighting function. Benefiting from this fine-tuning process, the pre-trained video diffusion model can effectively integrate event data, thereby enabling a high-fidelity synthesis of dynamic visual content.

## 4.2 INTER-FRAME TEMPORAL RESIDUAL GUIDANCE

Observing that each event is triggered when pixel intensity changes reach a specific threshold, a notable correlation exists between the accumulated events and the corresponding inter-frame residual at the same pixel. However, due to the differences in sensor sensitivities, gamma correction or other ISP algorithms, it is intractable to manually solve the inverse process that directly calculates the frame residual. Thus, we propose to leverage event representations to predict inter-frame residuals, employing these residuals as denoising guidance to balance reconstruction fidelity and diversity in the last $\tau$ steps.

Based on the reverse diffusion sampling process described in Eq. (3), we obtain the differential update of the latent as

$$\mathbf{X}^{t-1} = \mathbf{X}^t - \frac{\mathbf{X}^t - \mathbf{U}^t}{\sigma_t} \left(\sigma_t - \sigma_{t-1}\right). \tag{5}$$

As illustrated in Fig. 2, we initially train an off-the-shelf ResNet (He et al., 2016) model to map the event representation $\mathbf{E}$ to the inter-frame residual $\mathbf{R}$. Subsequently, at step $t$ ($t \leq \tau$) of the reverse diffusion process, the estimated clean latent $\mathbf{U}^t$ — derived from $\mathbf{X}^t$ — is fed into the autoencoder's decoder $\mathcal{D}$ to produce the estimated clean frame $\mathbf{F} = \mathcal{D}(\mathbf{U^t})$. The residual between each frame in

$\mathbf{F}$ and its preceding frame is calculated to obtain $\Delta\mathbf{F}$. We then compute the inter-frame residual loss function, which is formulated based on the L1 distance between the predicted inter-frame residual $\mathbf{R}$ and the residual derived from the estimated clean frame $\Delta\mathbf{F}$ in pixel space, i.e.,

$$\mathcal{L}_{\text{residual}}(\mathbf{U}^t) = |\Delta\mathbf{F} - \mathbf{R}| = \left|\Delta\mathcal{D}(\mathbf{U^t}) - \mathbf{R}\right|. \tag{6}$$

This loss function is optimized via the gradient descent algorithm to update the estimated clean latent $\mathbf{U}_t$ at each sampling step $t$:

$$\bar{\mathbf{U}}^t = \mathbf{U}^t - s\nabla_{\mathbf{U}^t}\mathcal{L}_{\text{residual}}(\mathbf{U}^t), \tag{7}$$

where $s$ is the coefficient that controls the strength of the guidance. Moreover, in the following Proposition, we theoretically validate that our regularization in Eq. (7) does not degrade generation quality and minimize such regularize can indeed improve generation quality by minimizing the potential error upper-boundary.

**Proposition 1.** *The gradient term $\nabla_{\mathbf{U}^t}\mathcal{L}_{residual}(\mathbf{U}^t)$ derived from the inter-frame residual guidance lies in the tangent space $T_{\mathbf{U}^t}\mathcal{M}$ of the data manifold $\mathcal{M}$ learned by the diffusion model. Then, we have the following characteristics.*

- *It ensures the updated latent $\bar{\mathbf{U}}^t = \mathbf{U}^t - s\nabla_{\mathbf{U}^t}\mathcal{L}_{residual}$ remains on $\mathcal{M}$.*
- *The reconstruction error is bounded by $\frac{L\kappa}{C}\mathcal{L}_{residual} + \frac{F\varepsilon}{C}$.*

*Proof.* See Appendix **??**. $\qquad\square$

By substituting the original $\mathbf{U}_t$ in the reverse diffusion Eq. (5) with the refined clean latent expectation term, the sampling process is guided as:

$$\mathbf{X}^{t-1} = \mathbf{X}^t - \frac{\mathbf{X}^t - \bar{\mathbf{U}}^t}{\sigma_t}\left(\sigma_t - \sigma_{t-1}\right). \tag{8}$$

### 4.3 Adaptation to Video Frame Interpolation and Prediction

Through the training on the event-based video frame reconstruction task, our approach is expected to acquire a powerful generative capability to map sparse, asynchronous event data to realistic and continuous video frames. Exploiting this ability, we extend our method to video frame interpolation and prediction without additional fine-tuning. Specifically, for video frame interpolation (or prediction), by leveraging the prior information from the first and last reference frames (or solely the first frame), we modulate the score function to theoretically reformulate the reverse diffusion sampling during the inference phase. This revised formulation then guides the video diffusion model to reconstruct the intermediate (or subsequent) frames with enhanced temporal consistency and visual fidelity.

Here, we take video frame interpolation as an example to illustrate our approach. Given the clean latents $\mathcal{E}(\mathbf{V}_0)$ and $\mathcal{E}(\mathbf{V}_{F-1})$, obtained by feeding the first and last video frames into the autoencoder's encoder, we first compute the deviations between them and the corresponding intermediate estimations $\mathbf{U}_0^t$ and $\mathbf{U}_{F-1}^t$:

---

**Algorithm 1** Reverse Diffusion Sampling for Video Frame Interpolation and Prediction

---

1: **Input:** clean latents $\{\mathcal{E}(\mathbf{V}_0), \mathcal{E}(\mathbf{V}_{F-1})\}$ (resp. $\{\mathcal{E}(\mathbf{V}_0)\}$) for the task of video frame interpolation (resp. prediction), diffusion U-Net $\mathcal{X}_\theta(\cdot)$, encoded event representation $\mathcal{E}(\mathbf{E})$.
2: initialize $\mathbf{X}^T \sim \mathcal{N}(\mathbf{0}, \sigma_t\mathbf{I})$.
3: **for** $t = T, \dots, 1$ **do**
4:     forward $\mathbf{U}^t = \mathcal{X}_\theta(\mathbf{X}^t, \mathcal{E}(\mathbf{E}), t)$.
5:     **for** $i = 0, \dots, F-1$ **do**
6:         **if** *video frame interpolation* **then**
7:             calculate $\tilde{\mathbf{U}}^t$ via Eq. (10)
8:         **else if** *video frame prediction* **then**
9:             calculate $\tilde{\mathbf{U}}^t$ via Eq. (12)
10:         **end if**
11:         reverse $\mathbf{X}^t$ to $\mathbf{X}^{t-1}$ via Eq. (11)
12:     **end for**
13: **end for**
14: **return** reconstructed latent $\mathbf{X}^0$.

---

$$\mathbf{D}_0^t = \mathcal{E}(\mathbf{V}_0) - \mathbf{U}_0^t, \ \mathbf{D}_{F-1}^t = \mathcal{E}(\mathbf{V}_{F-1}) - \mathbf{U}_{F-1}^t. \tag{9}$$

Since these deviations are highly correlated with the discrepancy between the estimated clean latents and the provided prior information, an effective score function can be designed by utilizing $\mathbf{D}_0^t$ and $\mathbf{D}_{F-1}^t$ to modulate the estimated latent representation $\mathbf{U}_i^t$ for $0 \leq i \leq F-1$. Thus, the score function of the reverse diffusion sampling is formulated:

$$\tilde{\mathbf{U}}_i^t = \alpha(t)\left[\frac{(\mathbf{D}_0^t + \mathbf{U}_i^t) + (\mathbf{D}_{F-1}^t + \mathbf{U}_i^t)}{2}\right] + [1 - \alpha(t)]\mathbf{U}_i^t, \tag{10}$$

Table 1: Quantitative comparison with state-of-the-art event-based video frame reconstruction methods on real-world and synthetic datasets. The best result in each column is highlighted in **bold**.

| Method | Real-World | | | Synthetic | | |
|---|---|---|---|---|---|---|
| | MSE ↓ | SSIM ↑ | LPIPS ↓ | MSE ↓ | SSIM ↑ | LPIPS ↓ |
| E2VID (Rebecq et al., 2019) | 0.1275 | 0.4200 | 0.6210 | 0.0678 | 0.5040 | 0.5420 |
| FireNet (Scheerlinck et al., 2020) | 0.1210 | 0.4110 | 0.6300 | 0.0620 | 0.5560 | 0.5220 |
| E2VID+ (Stoffregen et al., 2020) | 0.0650 | 0.4390 | 0.6180 | 0.0550 | 0.5130 | 0.5430 |
| FireNet+ (Stoffregen et al., 2020) | 0.0737 | 0.3870 | 0.6620 | 0.0581 | 0.4540 | 0.5790 |
| ETNet (Weng et al., 2021) | 0.0849 | 0.4230 | 0.6440 | 0.0522 | 0.5110 | 0.5480 |
| SSL-E2VID (Paredes-Vallés & De Croon, 2021) | 0.1008 | 0.3660 | 0.6260 | 0.0694 | 0.4980 | 0.6510 |
| SPADE-E2VID (Cadena et al., 2021) | 0.0727 | 0.4330 | 0.5990 | 0.0992 | 0.4500 | 0.6340 |
| CUBE (Zhao et al., 2024) | 0.0851 | 0.3640 | 0.6900 | 0.1437 | 0.1920 | 0.7970 |
| HyperE2VID (Ercan et al., 2024) | 0.0632 | 0.4770 | **0.5620** | 0.0727 | 0.3860 | 0.6320 |
| **UniE2F (Ours)** | **0.0612** | **0.4990** | 0.6740 | **0.0167** | **0.7100** | **0.3940** |

where the weighting coefficient $\alpha(t) = 1 - e^{-\sigma_t}$ balances the influence of the deviation correction. To guide the reverse sampling of the video diffusion model, we replace $\mathbf{U}^t$ in Eq. (5) with the optimized clean latent expectation term $\tilde{\mathbf{U}}_i^t$:

$$\mathbf{X}^{t-1} = \mathbf{X}^t - \frac{\mathbf{X}^t - \tilde{\mathbf{U}}^t}{\sigma_t}(\sigma_t - \sigma_{t-1}). \tag{11}$$

It is important to note that for video frame prediction, only $\mathcal{E}(\mathbf{V}_0)$ is available. In this case, the score function is modified into

$$\tilde{\mathbf{U}}_i^t = \alpha(t)(\mathbf{D}_0^t + \mathbf{U}_i^t) + [1 - \alpha(t)]\mathbf{U}_i^t. \tag{12}$$

Alg. 1 summarizes the reverse diffusion sampling process.

## 5 EXPERIMENT

### 5.1 EXPERIMENT SETTINGS

**Dataset.** The training set is generated by synthesizing event–frame pairs from real-world videos. Specifically, 1,800 sequences (400–500 frames each) are drawn from TrackingNet (Muller et al., 2018), and event stream between consecutive frames is simulated with DVS-Voltmeter (Lin et al., 2022). For evaluation, we construct a synthetic test set of 212 sequences from TrackingNet (Muller et al., 2018) and a real-world test set of 107 sequences from HS-ERGB (Tulyakov et al., 2021), each containing 12 frames. More details of the dataset are provided in Appendix B.

**Implementation Details.** All the experiments is conducted on an NVIDIA RTX A6000 GPU. In our experiments, the denoising U-Net architecture is initialized using the pre-trained weights of SVD model. During training, we selectively fine-tune the temporal transformer blocks while keeping all other parameters frozen, ensuring effective adaptation to the event-based domain. The network is optimized for 450,000 iterations over a period of 7 days using the AdamW optimizer (Loshchilov & Hutter, 2019) with a learning rate of $1 \times 10^{-5}$. In both training and inference stage we set $F = 12$ and employ 30 diffusion steps for the inference phase, with our event-based inter-frame residual guidance applied during the last 10 steps. Besides, we illustrate the training strategy for the inter-frame residual estimator in Appendix C. In terms of inference latency, reconstructing a sequence of 12 RGB frames with the resolution of $448 \times 320$ takes about 48 seconds. To quantitatively evaluate the reconstruction quality, we employ three widely used metrics: MSE (lower is better), SSIM (Wang et al., 2004) (higher is better), and LPIPS (Zhang et al., 2018) (lower is better).

### 5.2 RESULTS OF EVENT-BASED FRAME RECONSTRUCTION

We compare our method against various event-based reconstruction approaches, employing their official implementations and pre-trained parameters to ensure a fair evaluation. Notably, since the methods in (Liang et al., 2023), (Zhu et al., 2024c), and (Chen et al., 2024) are not open-sourced, and therefore they could not be included in our comparison. The quantitative results, as summarized in Table 1, reveal that our approach consistently outperforms the compared methods across multiple metrics. Specifically, our method outperforms state-of-the-art approaches, achieving the lowest MSE of 0.0612 and the highest SSIM of 0.4990 on the real-world dataset. Similar advantages are observed on the synthetic dataset, where our approach yields significant performance gains across the same evaluation metrics. Qualitative comparisons illustrated in Figure 3 reveal that our

Figure 3: Visual comparison of event-based video frame reconstruction results on synthetic (1st and 2nd rows) and real-world (3rd and 4th rows) datasets. The event data is visualized as polarity map.

Table 2: Quantitative comparison for video frame interpolation (VFI) and video frame prediction (VFP). The best result in each column is highlighted in **bold**. Here, "†" denotes the model with pretrained weight and "*" denotes the model retrained on our synthetic dataset.

| Method | Mode | Synthetic | | | Real-World | | |
|---|---|---|---|---|---|---|---|
| | | MSE ↓ | SSIM ↑ | LPIPS ↓ | MSE ↓ | SSIM ↑ | LPIPS ↓ |
| CBMNet† (Kim et al., 2023) | VFI-4× | 0.0250 | 0.5860 | **0.3030** | 0.0032 | 0.8230 | **0.2540** |
| CBMNet* (Kim et al., 2023) | VFI-4× | 0.0174 | 0.6320 | 0.3550 | **0.0023** | **0.8360** | 0.2810 |
| TimeLens-XL† (Ma et al., 2024) | VFI-4× | 0.0321 | 0.5350 | 0.3270 | 0.0080 | 0.7270 | 0.2920 |
| TimeLens-XL* (Ma et al., 2024) | VFI-4× | 0.0291 | 0.5480 | 0.3160 | 0.0078 | 0.7270 | 0.2900 |
| **UniE2F (Ours)** | VFI-4× | **0.0063** | **0.7340** | 0.3210 | 0.0041 | 0.6770 | 0.4310 |
| CBMNet† (Kim et al., 2023) | VFI-11× | 0.0491 | 0.4040 | 0.4580 | 0.0892 | 0.5120 | 0.5800 |
| CBMNet* (Kim et al., 2023) | VFI-11× | 0.0392 | 0.4530 | 0.5270 | 0.0063 | **0.7500** | 0.4000 |
| RE-VDM† (Chen et al., 2025) | VFI-11× | 0.0503 | 0.4180 | 0.4130 | **0.0057** | 0.7330 | **0.3480** |
| **UniE2F (Ours)** | VFI-11× | **0.0072** | **0.7400** | **0.3200** | 0.0058 | 0.6500 | 0.4000 |
| Zhu et al. (2024b)† | VFP | 0.0184 | 0.6140 | 0.3960 | **0.0077** | **0.6620** | **0.3400** |
| **UniE2F (Ours)** | VFP | **0.0093** | **0.7100** | **0.3470** | 0.0100 | 0.5940 | 0.4110 |

UniE2F, leveraging a large-scale generative prior along with inter-frame residual guidance, achieves reconstruction with improved color fidelity and fewer artifacts. In contrast, the compared methods, trained on grayscale images, tend to produce monochromatic outputs inconsistent with real-world color scenes, suffer from significant detail loss, and exhibit obvious artifacts.

## 5.3 RESULTS OF VIDEO FRAME INTERPOLATION AND PREDICTION

Here, we conduct experiments to evaluate the zero-shot capability of our UniE2F on the event-based video frame interpolation and prediction tasks. Specifically, we define three tasks: VFI-4×, VFI-11×, and VFP, which respectively increase the frame rate to 4 times and 11 times the original, and predict subsequent frames given the first frame. Following the above setup, we compare our method with several representative baselines for interpolation using their official pretrained weights or retraining them on our synthetic training set. Since TimeTracker (Liu et al., 2025) has not been open-sourced, its weights and results cannot be obtained and are therefore omitted. More detailed descriptions of the experimental setting are provided in Appendix D.

The quantitative and qualitative results are presented in Table 2 and Figure 4, respectively. On the synthetic dataset, UniE2F delivers a clear advantage in both short-range and long-range interpolation as well as prediction, consistently outperforming all pretrained and retrained methods. On the real-world datasets, UniE2F shows relatively weaker quantitative performance compared to retrained baselines, which is mainly attributed to the domain gap between synthetic training and real-world event distributions. Nevertheless, considering that our model operates in a strict zero-shot setting without any fine-tuning on real-world data, its ability to remain competitive on unseen real-world interpolation and prediction tasks is highly encouraging. Moreover, beyond numerical scores, UniE2F preserves motion dynamics while maintaining high-fidelity color, texture, and structural details, whereas outputs of other methods exhibit artifacts and degradation due to the lack of

| First Frame | UniE2F (VFP) | UniE2F (VFI) | Zhu et al. (VFP) | CBMNet (VFI) | Ground Truth | UniE2F (VFP) | UniE2F (VFI) | Zhu et al. (VFP) | CBMNet (VFI) | Ground Truth |

Figure 4: Visual comparison on synthetic dataset: UniE2F is shown in both video frame interpolation (VFI) and video frame prediction (VFP) modes, while CBMNet is shown under the VFI setting. Note that the frames highlighted with purple borders denote the given frames, whereas frames highlighted with blue borders denote the predicted frames.

generative priors. Remarkably, these zero-shot results are obtained **without any fine-tuning on interpolation or prediction datasets**, which confirms that UniE2Fprovides a unified and flexible framework that not only delivers excellent performance in event-driven frame reconstruction but also delivers outstanding zero-shot performance on both interpolation and prediction tasks, highlighting its generalization strength and practical utility.

## 5.4 ABLATION STUDY

We perform ablation studies on the synthetic dataset to assess the effectiveness of the proposed method. Additional and more comprehensive comparisons are provided in Appendix F.

**Guidance Strength.** We further investigate the optimal guidance strength strategy by evaluating four scheduling approaches under the experimental setup where the guidance is applied exclusively during the final 10 sampling steps. Specifically, we design four scheduling strategies over the designated steps: (**1**) a baseline configuration with no guidance, i.e., the guidance strength is set to 0; (**2**) maintaining a constant guidance strength of 0.1, (**3**) linearly decreasing the

Table 3: Comparison of different guidance strength strategies.

| Guidance Strength | Synthetic | | |
|---|---|---|---|
| | MSE ↓ | SSIM ↑ | LPIPS ↓ |
| 0.0 → 0.0 | 0.0191 | 0.6880 | 0.4020 |
| 0.1 → 0.1 | 0.0234 | 0.6610 | 0.4190 |
| 0.0 → 0.1 | 0.0234 | 0.6540 | 0.4210 |
| 0.1 → 0.0 | **0.0167** | **0.7100** | **0.3940** |

guidance strength from 0.1 to 0, and (**4**) linearly increasing the guidance strength from 0 to 0.1. The comparison results in Table 3 show that the linearly decreasing schedule (0.1 → 0.0) achieves the best performance compared with other configurations. The results indicate that applying stronger guidance early in the reverse diffusion process helps to enforce accurate inter-frame residual alignment, while gradually reducing the guidance allows the model's generative prior to effectively refine finer details and prevent over-constraining the reconstruction. This progressive relaxation appears critical for balancing reconstruction fidelity with visual diversity.

**Guidance Mode.** Since SVD model conducts the denoising process in the latent space, a direct strategy to enhance inter-frame consistency is to impose explicit constraints on the residuals between adjacent frames within this space. To this end, we trained a ResNet with the aim of predicting inter-frame residuals within latent representation. As shown in Figure 5 and Table 4, the results of latent-level guidance are inferior to frame-level guidance results. Moreover, the images generated with frame-level guidance exhibit higher fidelity and richer details, producing sharper structures and more natural textures In contrast, the outputs under latent-level guidance often suffer from noticeable distortions and artifacts, leading to degraded perceptual quality. The performance gap can be attributed to Gaussian distribution

| Frame | Latent | Ground Truth |

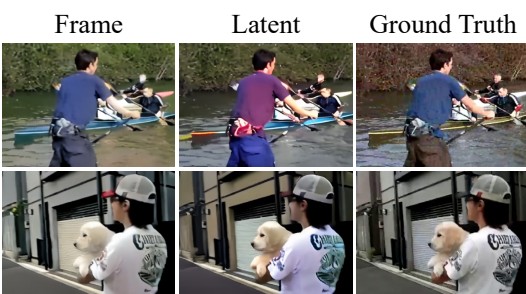

Figure 5: Visual comparison of guidance modes.

Table 4: Comparison of different guidance modes.

| Guidance Mode | Synthetic | | |
|---|---|---|---|
| | MSE ↓ | SSIM ↑ | LPIPS ↓ |
| Latent | 0.0197 | 0.6920 | 0.4060 |
| Frame | **0.0167** | **0.7100** | **0.3940** |

inherent in the latent inter-frame residuals, making it particularly challenging for networks trained with MAE loss to accurately model the precise variations.

**Weighting Coefficient.** In video frame interpolation tasks, we compared two weight-scheduling strategies: 1) Nonlinear schedule, as defined by the formula in the main text; 2) Linear schedule, i.e. linearly decreasing (increasing) from 1.0 (0.0) to 0.0 (1.0). The quantitative results are presented in Table 5. As shown in the table, the nonlinear schedule generally achieves lower MSE and LPIPS scores while maintaining competitive SSIM, indicating a more effective fusion of reference-frame information compared to the linear schedule. In particular, the "Nonlinear $1.00 \rightarrow 0.00$" configuration

Table 5: Quantitative comparison of different weight-scheduling strategies for video frame interpolation (VFI), evaluated using MSE, SSIM, and LPIPS.

| Mode | Weighting Coefficient | Synthetic | | |
|---|---|---|---|---|
| | | MSE ↓ | SSIM ↑ | LPIPS ↓ |
| Linear | $0.50 \rightarrow 0.50$ | 0.0080 | 0.7110 | 0.3460 |
| Linear | $0.00 \rightarrow 1.00$ | 0.0082 | 0.7030 | 0.3540 |
| Linear | $1.00 \rightarrow 0.00$ | 0.0076 | **0.7450** | 0.3220 |
| Nonlinear | $0.00 \rightarrow 1.00$ | 0.0090 | 0.6960 | 0.3630 |
| Nonlinear | $1.00 \rightarrow 0.00$ | **0.0072** | 0.7400 | **0.3200** |

yields the best overall performance , consistently reducing distortion and perceptual error. Therefore, comprehensively considering MSE, SSIM, and LPIPS, we adopt "Nonlinear $1.00 \rightarrow 0.00$" as the default setting.

**Robustness to Event Noise.** We follow the setting in SVD model, where in the inference stage Gaussian noise with standard deviation 0.02 is injected into the 3-channel event representation. As a reference, we regard this original noise injection scheme as the *Baseline*. To further evaluate the robustness of our network to event noise, we introduce a noise-level coefficient $\alpha$ to scale the noise strength relative to the standard deviation of the event representation $E$: $E^{\mathrm{noisy}} = E_{f,c,y,x} + \mathcal{N}\left(0, \ (\alpha \times \sigma_E)^2\right), \quad \sigma_E = \mathrm{std}(E)$. By varying $\alpha$, we systematically control the injected noise magnitude. From Table 6, we observe

Table 6: Quantitative comparison of different noise-level coefficient, evaluated using MSE, SSIM, and LPIPS. The results demonstrates that our method exhibits strong robustness to event noise.

| Noise-level Coefficient | Synthetic | | |
|---|---|---|---|
| | MSE ↓ | SSIM ↑ | LPIPS ↓ |
| 1.0 | 0.0193 | 0.6690 | 0.4280 |
| 0.5 | 0.0177 | 0.6930 | 0.4050 |
| 0.1 | 0.0168 | 0.7080 | 0.3950 |
| Baseline | **0.0167** | **0.7100** | **0.3940** |

that even under strong noise ($\alpha = 1.0$), the reconstruction quality only slightly degrades: MSE rises from 0.0167 to 0.0193, SSIM drops from 0.7100 to 0.6690, and LPIPS increases from 0.3940 to 0.4280. These marginal changes demonstrate that our method maintains strong robustness to event noise.

**Max Guidance Strength.** In the ablation experiments for the video frame reconstruction task, we expanded the maximum guidance strength from 0.1 to larger values such as 0.2 and 0.3, and compared MSE, SSIM and LPIPS (see Table 7). The results show that as the coefficient grows, MSE increases, SSIM drops, and LPIPS rises—indicating that overly strong residual guidance causes the reconstruction to depend too heavily on the event–frame residual. Because the mapping

Table 7: Quantitative comparison of different maximum guidance strength, evaluated using MSE, SSIM, and LPIPS.

| Maximum Guidance Strength | Synthetic | | |
|---|---|---|---|
| | MSE ↓ | SSIM ↑ | LPIPS ↓ |
| 0.3 | 0.0176 | 0.6970 | 0.4030 |
| 0.2 | 0.0170 | 0.7070 | 0.3960 |
| 0.1 | **0.0167** | **0.7100** | **0.3940** |

from events to RGB involves nonlinear steps (e.g. gamma correction, ISP pipeline) and is not one-to-one, amplifying this loss instead introduces artifacts and degrades visual quality.

# 6 CONCLUSION

We have presented a novel event-based video frame reconstruction approach by fine-tuning a pre-trained SVD model with event data as conditional inputs. Leveraging the powerful generative prior from the pre-trained video diffusion model, our method significantly enhances the fidelity and realism of reconstructed frames, especially in complex dynamic scenes. The introduction of event-based inter-frame residual guidance further improves the accuracy while maintaining diversity in the reconstruction. Our unified framework is versatile, not only excelling in video frame reconstruction but also extending seamlessly to tasks such as interpolation and prediction. Experimental results on both synthetic and real-world datasets validate the effectiveness of our approach, demonstrating improved performance compared to existing methods across multiple evaluation metrics.

## ETHICS STATEMENT

This research does not involve human subjects, animal experiments, or sensitive personal data. The datasets used are either publicly available or will be released by the authors upon acceptance of this paper. We do not anticipate any negative ethical, societal, or environmental impacts arising from this work.

## REPRODUCIBILITY STATEMENT

We are committed to the reproducibility of our research. Upon acceptance, we will release the source code and datasets associated with this work. Detailed descriptions of dataset construction are provided in Section 5.1 of the main paper and Appendix B. Information on model architectures, training procedures, and hyperparameters is presented in Section 5.1 of the main paper and Appendix C. The proof of Proposition 1 is included in Section **??**. These resources and descriptions are intended to enable independent verification and replication of our results.

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

APPENDIX CONTENTS

## A   PROOFS

### A.1   GRADIENT ALIGNMENT WITH THE TANGENT SPACE & MANIFOLD-PRESERVED SAMPLING

Proof of **Gradient Alignment with the Tangent Space** is based on the following **assumptions**:

- the data manifold $\mathcal{M}$ is a smooth embedded submanifold of $\mathbb{R}^n$;
- the decoder $\mathcal{D} : \mathcal{M} \to \mathcal{P}$ (pixel space) is differentiable with a full-rank Jacobian $J_{\mathcal{D}}$, corresponding to the state $\mathbf{U}^t$.

Due to that our introduced residual loss is defined in pixel space as

$$\mathcal{L}_{\text{residual}} = \|\Delta \mathcal{D}(\mathbf{U}^t) - \mathbf{R}\|_1, \tag{13}$$

where $\Delta \mathcal{D}(\mathbf{U}^t) = \mathcal{D}(\mathbf{U}^t_{1:F}) - \mathcal{D}(\mathbf{U}^t_{0:F-1})$. Then, the gradient in pixel space is:

$$\nabla_{\Delta \mathbf{F}} \mathcal{L}_{\text{residual}} = \text{sign}(\Delta \mathbf{F} - \mathbf{R}). \tag{14}$$

By the chain rule, the gradient in latent space is:

$$\nabla_{\mathbf{U}^t} \mathcal{L}_{\text{residual}} = J_{\mathcal{D}}^\top(\mathbf{U}^t) \cdot \nabla_{\Delta \mathbf{F}} \mathcal{L}_{\text{residual}}, \tag{15}$$

where $J_{\mathcal{D}}$ is the Jacobian of $\mathcal{D}$. Since $\mathcal{D}$ maps $\mathcal{M}$ to $\mathcal{P}$, $J_{\mathcal{D}}$ acts as a linear map from $T_{\mathbf{U}^t}\mathcal{M}$ to $T_{\mathcal{D}(\mathbf{U}^t)}\mathcal{P}$. Thus, $\nabla_{\mathbf{U}^t} \mathcal{L}_{\text{residual}} \in T_{\mathbf{U}^t}\mathcal{M}$.

Then, we prove that our proposed updating is **manifold-preserved**. The part is based on the following assumption: The step size $s$ satisfies $s \cdot \|\nabla_{\mathbf{U}^t} \mathcal{L}_{\text{residual}}\| < \delta$, where $\delta$ is the injectivity radius of $\mathcal{M}$ at $\mathbf{U}^t$.

The exponential map $\exp_{\mathbf{U}^t} : T_{\mathbf{U}^t}\mathcal{M} \to \mathcal{M}$ defines a local diffeomorphism (a mapping between a group of manifolds with similar structure) (Chung et al., 2023) within the radius $\delta$. Then, we can express the update via the exponential map:

$$\tilde{\mathbf{U}}^t = \exp_{\mathbf{U}^t}\left(-s\nabla_{\mathbf{U}^t}\mathcal{L}_{\text{residual}}\right). \tag{16}$$

For $s \cdot \|\nabla_{\mathbf{U}^t}\mathcal{L}_{\text{residual}}\| < \delta$, $\exp_{\mathbf{U}^t}$ ensures $\tilde{\mathbf{U}}^t \in \mathcal{M}$. Thus, for small $s$, the update approximates the Euclidean step:

$$\tilde{\mathbf{U}}^t \approx \mathbf{U}^t - s\nabla_{\mathbf{U}^t}\mathcal{L}_{\text{residual}} + \mathcal{O}(s^2\|\nabla_{\mathbf{U}^t}\mathcal{L}_{\text{residual}}\|^2), \tag{17}$$

with higher-order terms bounded by the sectional curvature of $\mathcal{M}$. Due to that we have proved that our updating gradient is in the tangent space of diffusion manifold, thus, .

## A.2 RECONSTRUCTION ERROR IS BOUNDED

We first reclaim the following **notions**. Let $\mathbf{V} = \{\mathbf{V}_0, \mathbf{V}_1, ..., \mathbf{V}_{F-1}\}$ denote the ground truth video frames. Let $\mathbf{F} = \{\mathbf{F}_0, \mathbf{F}_1, ..., \mathbf{F}_{F-1}\}$ denote the reconstructed frames, where $\mathbf{F}_k = \mathcal{D}(\mathbf{U}_k)$. Moreover, we also claim the following assumptions, which is the base for the following proof.

- The event-derived residual $\mathbf{R}_k$ approximates $C \cdot \Delta\mathbf{V}_k$, where $C$ is the event camera's contrast threshold, and $\Delta\mathbf{V}_k = \mathbf{V}_{k+1} - \mathbf{V}_k$.
- The decoder $\mathcal{D} : \mathcal{M} \to \mathcal{P}$ is $L$-Lipschitz smooth, i.e., $\|\mathcal{D}(\mathbf{U}) - \mathcal{D}(\mathbf{U}')\| \leq L\|\mathbf{U} - \mathbf{U}'\|$.
- The Jacobian $J_{\mathcal{D}}$ of $\mathcal{D}$ is full-rank and bounded: $\|J_{\mathcal{D}}\| \leq L$, $\|J_{\mathcal{D}}^{-1}\| \leq \kappa$, where $\kappa$ is the condition number.

Note that we have the following residual loss to train the encoder. It enforces alignment between reconstructed and event-derived residuals:

$$\mathcal{L}_{\text{residual}} = \sum_{k=0}^{F-2} \|\Delta\mathbf{F}_k - \mathbf{R}_k\|_1, \tag{18}$$

where $\Delta\mathbf{F}_k = \mathbf{F}_{k+1} - \mathbf{F}_k$. From the event camera model:

$$\mathbf{R}_k = C \cdot \Delta\mathbf{V}_k + \boldsymbol{\epsilon}_k, \quad \|\boldsymbol{\epsilon}_k\|_1 \leq \varepsilon, \tag{19}$$

where $\boldsymbol{\epsilon}_k$ captures event noise and threshold approximation errors.

We first to find the correlation with the residual loss and frame reconstruction error. The reconstruction error for frame $k$ is:

$$\|\mathbf{F}_k - \mathbf{V}_k\|_1 \leq \sum_{i=0}^{k-1} \|\Delta\mathbf{F}_i - \Delta\mathbf{V}_i\|_1. \tag{20}$$

By using the triangle inequality, we have

$$\|\Delta\mathbf{F}_i - \Delta\mathbf{V}_i\|_1 \leq \|\Delta\mathbf{F}_i - \mathbf{R}_i/C\|_1 + \|\mathbf{R}_i/C - \Delta\mathbf{V}_i\|_1. \tag{21}$$

Then, through substituting $\mathbf{R}_k = C \cdot \Delta\mathbf{V}_k + \boldsymbol{\epsilon}_k$ into aforementioned equation, we have

$$\|\Delta\mathbf{F}_i - \Delta\mathbf{V}_i\|_1 \leq \frac{1}{C}\|\Delta\mathbf{F}_i - \mathbf{R}_i\|_1 + \frac{1}{C}\|\boldsymbol{\epsilon}_i\|_1. \tag{22}$$

Summing over all frames:

$$\|\mathbf{F}_k - \mathbf{V}_k\|_1 \leq \frac{1}{C}\sum_{i=0}^{k-1}\left(\|\Delta\mathbf{F}_i - \mathbf{R}_i\|_1 + \varepsilon\right). \tag{23}$$

Then, we try to bound the latent space deviation. The residual loss gradient corrects the latent $\mathbf{U}$ via:

$$\tilde{\mathbf{U}}_k = \mathbf{U}_k - s\nabla_{\mathbf{U}_k}\mathcal{L}_{\text{residual}}. \tag{24}$$

Using the Lipschitz continuity of $\mathcal{D}$:

$$\|\Delta\mathbf{F}_k - \mathbf{R}_k\|_1 \leq L\|\Delta\mathbf{U}_k - \Delta\mathbf{U}_k^*\|_1, \tag{25}$$

where $\Delta\mathbf{U}_k^* = J_{\mathcal{D}}^{-1} \cdot \Delta\mathbf{V}_k$ is the ideal latent residual. From the Jacobian bound:

$$\|\Delta\mathbf{U}_k - \Delta\mathbf{U}_k^*\|_1 \leq \kappa\|\Delta\mathbf{F}_k - C \cdot \Delta\mathbf{V}_k\|_1. \tag{26}$$

Final Reconstruction Error Bound. Combining the above:

$$\|\mathbf{F} - \mathbf{V}\|_1 \leq \frac{1}{C}\sum_{k=0}^{F-2}\left(L \cdot \kappa\|\Delta\mathbf{F}_k - C \cdot \Delta\mathbf{V}_k\|_1 + \varepsilon\right). \tag{27}$$

Substituting $\|\Delta\mathbf{F}_k - C \cdot \Delta\mathbf{V}_k\|_1 \leq \|\Delta\mathbf{F}_k - \mathbf{R}_k\|_1 + \|\mathbf{R}_k - C \cdot \Delta\mathbf{V}_k\|_1$: Then, we can draw the final conclusion that the total reconstruction error is bounded by:

$$\|\mathbf{F} - \mathbf{V}\|_1 \leq \underbrace{\frac{L\kappa}{C}\mathcal{L}_{\text{residual}}}_{\text{Controlled by gradient guidance}} + \underbrace{\frac{F\varepsilon}{C}}_{\text{Event model error}}. \tag{28}$$

Minimizing $\mathcal{L}_{\text{residual}}$ directly reduces the error, while the term $\frac{F\varepsilon}{C}$ is inherent to event data noise.

## B  DATASET

The training of the network relies on a sufficiently large collection of event sequences paired with corresponding ground-truth video frames. However, due to the limited availability of real-world datasets, we employ the event simulator to synthesize a large-scale training dataset from existing video data. Specifically, we select 1,800 video frame sequences which have large motion from TrackingNet (Muller et al., 2018)—a large-scale object tracking dataset that has an extensive collection of real-world scenarios captured from YouTube videos. Each sequence comprises approximately 400 to 500 frames. Subsequently, we employ the DVS-Voltmeter (Lin et al., 2022) to simulate event streams between consecutive frames, thereby creating synthetic event data. Unlike previous works that generate synthetic datasets from MS-COCO (Lin et al., 2014), which only contains globally homographic motion, our dataset is simulated from real-world videos and incorporates both globally homomorphic motion and locally independent motion. This diversity can enable the network to generalize effectively to the various camera movements and scene variations encountered in real-world scenarios.

To ensure an accurate assessment of the algorithm's performance in reconstructing video frames, we also construct a synthetic test set and a real-world test set. The synthetic test set is created by selecting 212 video frame sequences from TrackingNet (Muller et al., 2018), with each sequence containing 12 frames. In contrast, the real-world test set, consisting of video sequences (each with 12 frames), is collected from the High-Speed Events and RGB (HS-ERGB) (Tulyakov et al., 2021) dataset, which contains synchronized event data and RGB frames recorded from a hybrid camera system.

## C  TRAINING STRATEGY FOR THE INTER-FRAME RESIDUAL ESTIMATOR

First, we select event streams and their corresponding RGB video frames from the synthetic dataset's training set, and convert each event stream into a three-channel event representation; then, this

representation is fed into a ResNet backbone (identical to the standard ResNet except that its output layer is modified to predict pixel-level residuals between consecutive frames). During training, we supervise the network with the pixel-wise difference between consecutive video frames, $\mathbf{V}_{t+1} - \mathbf{V}_t$, and optimize it by minimizing the $L_1$ loss between the predicted and ground-truth residuals. The network is trained for 9,000 iterations with the Adam optimizer ($\beta_1 = 0.9$, $\beta_2 = 0.999$) at a fixed learning rate of $1 \times 10^{-4}$. This setup ensures the estimator effectively learns the mapping from event representations to inter-frame residuals, providing accurate priors for the subsequent residual guidance.

## D    EXPERIMENTAL SETTING FOR VIDEO FRAME INTERPOLATION AND PREDICTION

Given a sequence of twelve consecutive frames from the test set, the network takes the reference frames in the case of video frame interpolation (VFI) to reconstruct the intermediate frames, or the first frame in the case of video frame prediction (VFP) to generate the subsequent frames.

For the interpolation task, to ensure compatibility with different interpolation factors supported by various pretrained baselines, we introduce two temporal up-sampling settings during evaluation: VFI-4× and VFI-11×. Specifically, in the VFI-4× setting, we use frames at indices 0, 4, 8 as references and interpolate six intermediate frames (three intermediate frames between each pair. i.e., frames 1–3 for 0, 4 and 5–7 for 4, 8). In the VFI-11× setting, we interpolate ten frames between frames at indices 0, 11. For the prediction task, we use the first frame at index 0 to generate the subsequent eleven frames at indices from 1 to 11. After obtaining the intermediate or subsequent frames from the network output, we compare them with the corresponding ground-truth frames from the test set to compute MSE, SSIM, and LPIPS.

## E    DETAILED ILLUSTRATION OF UNIE2F

In Figure 6, we provide a detailed schematic of our framework. At step $t$, given event representations, we utilize a ResNet to predict the inter-frame residuals between consecutive frames. Then, these residuals are utilized to formulate the inter-frame residual loss $L_{residual}$, which is optimized via gradient descent algorithm to update the noisy latent. Note that $\mathcal{E}$ and $\mathcal{D}$ denote the encoder and decoder in stable video diffusion model.

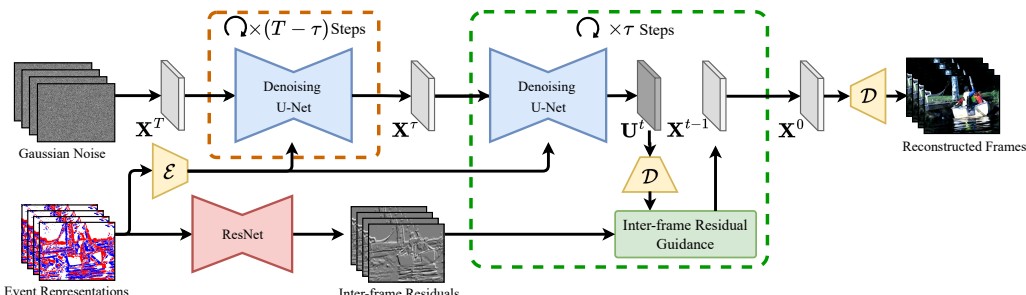

Figure 6: Illustration of our UniE2F.

## F    ADDITIONAL ABLATION STUDY

**Effectiveness of Inter-Frame Residual Guidance**  To demonstrate its effectiveness of the Inter-Frame Residual Guidance (IFRG), we have included a visual comparison in Figure 7. As shown, without IFRG (top row), the reconstructions exhibit noticeable blurring and structural distortion: the bars of the fence are wavy and over-smoothed, and the ground textures are largely washed out. With IFRG (middle row), edges and fine structures align much better with the ground truth (bottom row): the fence bars are straighter and more regular, the sand and background textures are more faithfully recovered. These improvements indicate that IFRG effectively constrains the reconstruction with inter-frame intensity changes, yielding frames that are both structurally more accurate to the ground truth.

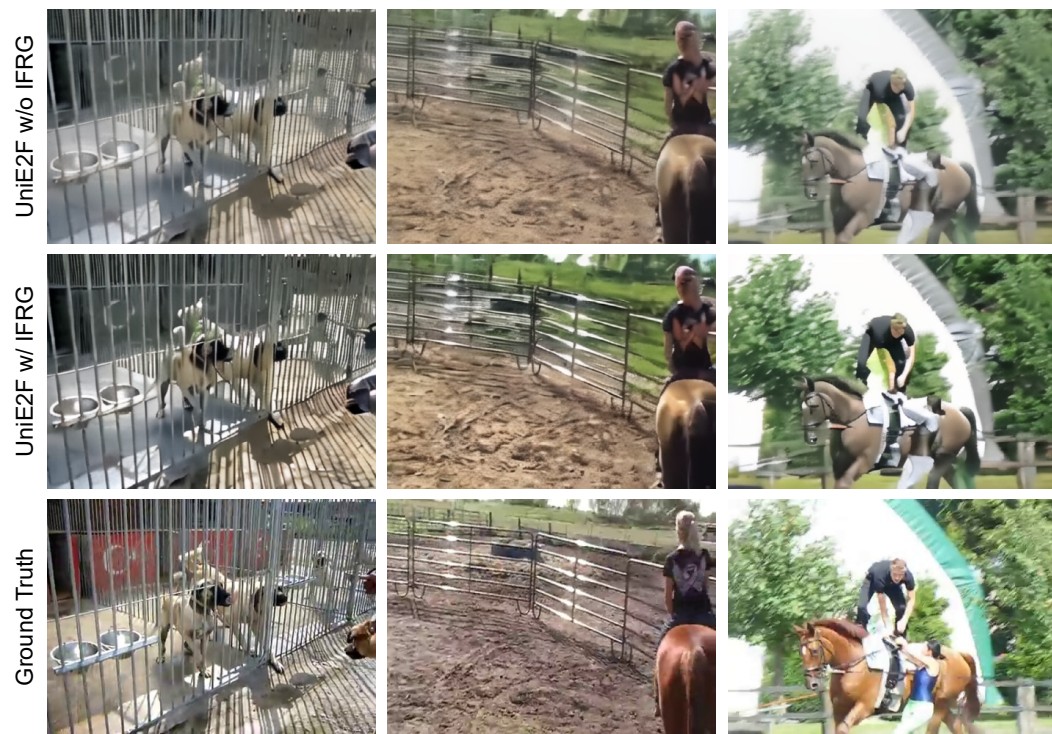

Figure 7: Effect of Inter-Frame Residual Guidance (IFRG). Compared with the baseline (top), adding IFRG (middle) yields sharper edges and more faithful textures that better match the ground truth (bottom).

Table 8: Quantitative comparison of UniE2F with and without event guidance for prediction and interpolation.

| Task | MSE ↓ | SSIM ↑ | LPIPS ↓ |
|---|---|---|---|
| Prediction (w/ Event) | **0.0093** | **0.7100** | **0.3470** |
| Prediction (w/o Event) | 0.1563 | 0.107 | 0.658 |
| Interpolation (w/ Event) | **0.0072** | **0.7400** | **0.3200** |
| Interpolation (w/o Event) | 0.1437 | 0.119 | 0.633 |

**Video Frame Interpolation/Prediction without Event.** To illustrate the contribution of event data, we conducted experiments to evaluate the performance of UniE2F with only image input in interpolation and prediction tasks, i.e., without using events as conditional guidance. The qualitative results on synthetic datasets are shown in Table 8 and Figure 8. As can be seen from the table and figure, UniE2F has learned during training to map event representations to RGB frames. When we remove the event input and, following SVD, condition the model only on RGB frames that lie in a different domain, the network effectively applies this learned event-to-RGB mapping to RGB-to-RGB conditioning instead. This mismatch causes strong artifacts and disturbed appearance patterns. This demonstrates the critical role that event data plays in providing precise constraints, enabling the model to reconstruct intermediate or subsequent frames by adjusting the score function, even without explicit training for interpolation and prediction tasks.

**Reconstruction under Sparse Event Streams.** To further analyze the limitations of event-based video reconstruction, we investigate the behavior of UniE2F and competing methods in scenarios with extremely sparse events. Figure 9 presents qualitative comparisons of reconstructed frames under such sparse-event conditions. As can be seen from the figure, when the event stream becomes highly sparse, all event-based frame reconstruction methods—including ours—can only reliably recover structures in the vicinity of pixels where events are triggered. Regions without event activity provide little or no informative signal, and therefore cannot be faithfully reconstructed by any of these approaches. This illustrates a fundamental limitation shared by event-only reconstruction

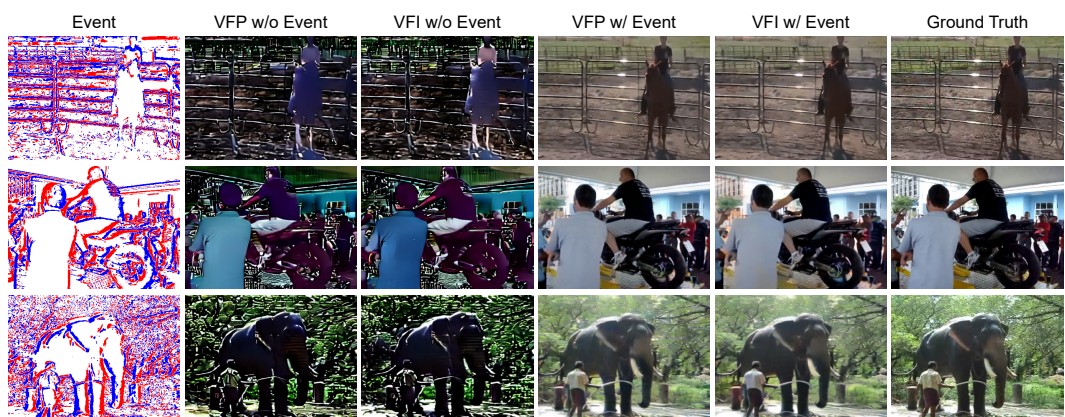

Figure 8: Video frame interpolation/prediction without events.

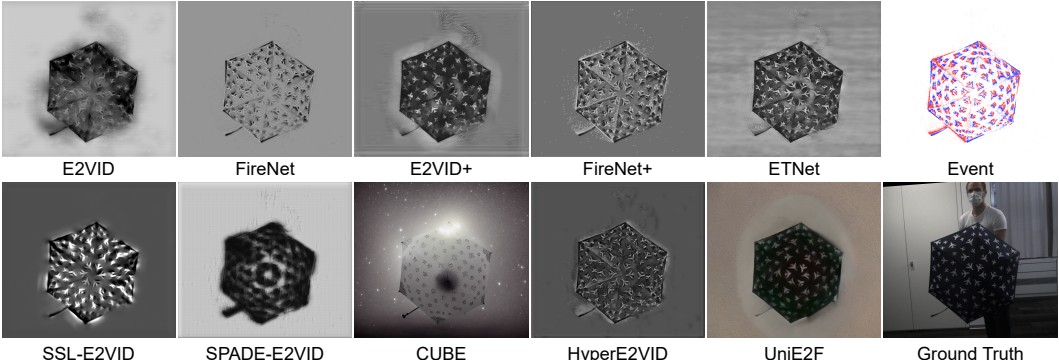

Figure 9: Reconstruction under sparse event streams. All event-based methods recover structures near event pixels, but fail to reconstruct regions without events.

Table 9: Comparison of computational overhead between our UniE2F and other methods, evaluated by reconstructing a sequence of 12 RGB frames at 448×320 resolution.

| Method | Computational Cost (TMACs) | Peak Memory Footprint (MB) |
|---|---|---|
| E2VID | 0.583 | 622 |
| FireNet | 0.064 | 462 |
| E2VID+ | 0.782 | 816 |
| FireNet+ | 0.064 | 436 |
| ETNet | 1.737 | 1074 |
| SSL-E2VID | 0.065 | 640 |
| SPADE-E2VID | 0.226 | 1096 |
| CUBE | 300.116 | 8215 |
| HyperE2VID | 0.060 | 1052 |
| **UniE2F (Ours)** | 245.364 | 46753 |

methods: in the absence of sufficient event observations, the model is unable to infer fine-grained details in event-free areas.

## G    COMPARISON OF COMPUTATIONAL OVERHEAD

We evaluated the efficiency of our UniE2Fand other methods on a single NVIDIA RTX A6000 GPU by reconstructing a sequence of 12 RGB frames at 448×320 resolution. The result is listed in Table 9. Although UniE2F's use of a pre-trained stable video diffusion model's generative prior incurs higher computational cost and GPU memory usage than non-diffusion methods, it provides

Table 10: Quantitative comparison of different maximum guidance strength, evaluated using MSE, SSIM, and LPIPS.

| Maximum Guidance Strength | Synthetic | | |
|---|---|---|---|
| | MSE ↓ | SSIM ↑ | LPIPS ↓ |
| 10.0 | 0.0964 | 0.2950 | 0.7070 |
| 5.0 | 0.0573 | 0.4140 | 0.6510 |
| 1.0 | 0.0228 | 0.6140 | 0.4770 |
| 0.5 | 0.0188 | 0.6710 | 0.4240 |
| 0.1 | **0.0167** | **0.7100** | **0.3940** |

Table 11: Quantitative comparison of different guidance strength strategies, evaluated using MSE, SSIM, and LPIPS.

| Guidance Strength Strategy | Synthetic | | |
|---|---|---|---|
| | MSE ↓ | SSIM ↑ | LPIPS ↓ |
| Constant | 0.0234 | 0.6610 | 0.4190 |
| Exponential | 0.0181 | 0.6960 | 0.3990 |
| Linear | **0.0167** | **0.7100** | **0.3940** |

a high-quality baseline for event-to-frame reconstruction. Moreover, the Stable Diffusion–based CUBE requires both user-provided text prompts and event data, and its reconstruction fails when only pure events are supplied. By contrast, our method still significantly outperforms CUBE in reconstruction fidelity and visual quality even without any textual prompts. In future work, building on the proposed UniE2F, we will investigate diffusion-model distillation, reducing the number of sampling steps, and network pruning to speed up inference for real-time applications.

## H    COMPARISON ON HQF, IJRR, AND MVSEC

We further compare UniE2F with existing methods on three real-world event camera datasets: HQF (Stoffregen et al., 2020), IJRR (Mueggler et al., 2017), and MVSEC (Zhu et al., 2018), which provide single-channel intensity frames as ground truth. As shown in  Figure 10, UniE2F can reconstruct frames with more realistic colors and clear details, making them closer to real scenes. In contrast, the other methods can only produce single-channel grayscale reconstructions, which lack color information and look less natural.

## I    EFFECT OF WEIGHTING FACTOR IN INTER-FRAME RESIDUAL

The value of the weighting factor $s$ controls the gradient update magnitude for the estimated clean latent $U_t$ by weighting the contribution of the inter-frame residual loss. To evaluate the effect of the weighting coefficient $s$ on performance, we conduct additional experiments where $s$ was set to 0.5, 1.0, and 10.0. The results of these experiments are presented in Table 10. The results indicate that $s = 0.1$ achieves the best performance across all three metrics (lowest MSE and LPIPS, highest SSIM).

## J    EFFECT OF GUIDANCE STRENGTH SCHEDULING

In the ablation study of main paper, we have investigated three different guidance strength strategies—linearly increasing, linearly decreasing, and constant guidance strengths—and the results show that the linearly decreasing schedule achieves the best performance. Here, to evaluate the robustness of this schedule, we introduce a non-linearly exponential decreasing strategy, where the guidance strength decreases from 0.1 to 0 following an exponential curve. The comparison results are presented in the Table 11. As shown, the linearly decreasing schedule still outperforms the exponential decreasing one.

## K    EFFECT OF VOXEL GRID VS. THREE-CHANNEL EVENT REPRESENTATION

We further investigate how the choice of event representation affects the performance of UniE2F by retraining our model using a voxel grid with three temporal bins as the conditioning input. The

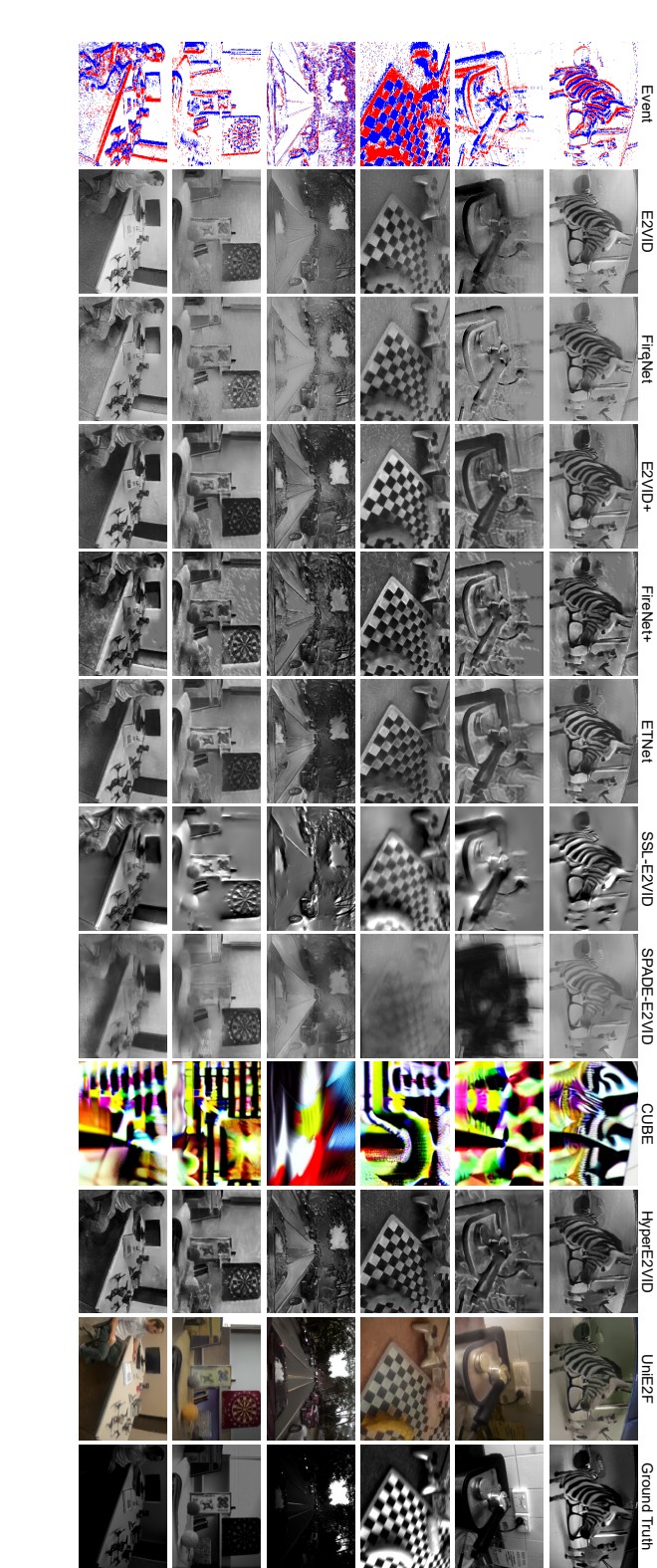

Figure 10: Qualitative Comparison on HQF (Stoffregen et al., 2020), IJRR (Mueggler et al., 2017), and MVSEC (Zhu et al., 2018).

Table 12: Quantitative comparison of UniE2F trained with three-channel event representation vs. three-bin voxel grid.

| Encoding | MSE ↓ | SSIM ↑ | LPIPS ↓ |
|---|---|---|---|
| Three-Bin Voxel Grid | 0.0172 | **0.7290** | 0.4060 |
| Three-Channel Event Representation | **0.0167** | 0.7100 | **0.3940** |

Table 13: Quantitative comparison of UniE2F trained on real-world dataset and synthetic dataset.

| Training Data | MSE ↓ | SSIM ↑ | LPIPS ↓ |
|---|---|---|---|
| Real-World | 0.1084 | 0.3080 | 0.7580 |
| Synthetic | **0.0167** | **0.7100** | **0.3940** |

Table 14: Quantitative results for long-sequence generation on synthetic dataset.

| Method | MSE ↓ | SSIM ↑ | LPIPS ↓ |
|---|---|---|---|
| SSL-E2VID (Paredes-Vallés & De Croon, 2021) | 0.0906 | 0.4240 | 0.6310 |
| SPADE-E2VID (Cadena et al., 2021) | 0.0682 | 0.5200 | 0.5680 |
| HyperE2VID (Ercan et al., 2024) | 0.0591 | 0.4960 | 0.5700 |
| **UniE2F** | **0.0271** | **0.6530** | **0.4470** |

quantitative comparison on synthetic dataset is reported in Table 12. As shown in the table, the performance difference between the voxel-grid representation and our three-channel event representation is very small, indicating that the specific form of these two reasonable encodings has a limited impact on the final results. This suggests that, in our setting, the data itself (i.e., the training distribution and supervision) is the primary factor, while the difference between these two representations is secondary.

## L  EFFECT OF TRAINING UNIE2F ON REAL-WORLD EVENT DATA

Here, we split the HS-ERGB (Tulyakov et al., 2021) real-world dataset into a training set and a test set, and trained UniE2F directly on its training split. The quantitative results on the test split are reported in Table 13. From the table, we observe that the limited size and diversity of HS-ERGB (Tulyakov et al., 2021) cause the model to overfit a few specific scenes when learning the event-to-frame mapping, which in turn weakens the generalization ability brought by large-scale pre-training. Therefore, in the main paper, we choose to train UniE2F on a more diverse synthetic dataset and evaluate it on real-world datasets.

## M  LONG-SEQUENCE EVENT-BASED VIDEO RECONSTRUCTION

To further evaluate our method on long sequences, we follow the same data collection setup as in the main paper. The long-sequence synthetic test set is built by sampling 200 video sequences from TrackingNet (Muller et al., 2018), each containing 23 frames. For UniE2F, we first perform event-to-frame reconstruction on the first 12 frames of each sequence, obtaining a 12-frame reconstructed sequence. Then, starting from the 12-th frame, we use the following events and apply our prediction mode to generate the subsequent 11 frames. We compare UniE2F with recent event-based video frame reconstruction methods on this setting. As shown in Table 14, our method achieves better performance across the reported metrics, indicating that UniE2F maintains stronger reconstruction quality and temporal stability even when the sequence length is increased.

## N  EVENT-DRIVEN INTERPOLATION AT ARBITRARY TIMESTAMPS

Here, we provide the experiment to evaluate the interpolation capability of our UniE2F at arbitrary timestamps. Concretely, we take a sequence of 12 frames. Concretely, we take a sequence of 12 frames $\{V_0, V_1, \ldots, V_{11}\}$ from the dataset and the 12 corresponding event groups $\{\mathcal{G}_0, \mathcal{G}_1, \ldots, \mathcal{G}_{11}\}$. Then, for the interpolation task, we **focus only on the frames between the first and last frames**, i.e., the 10 event groups $\{\mathcal{G}_1, \mathcal{G}_2, \ldots, \mathcal{G}_{10}\}$ associated with $\{V_1, V_2, \ldots, V_{10}\}$. These 10 intermediate

First Frame    Intermediate Frame #1    Intermediate Frame #2    Last Frame

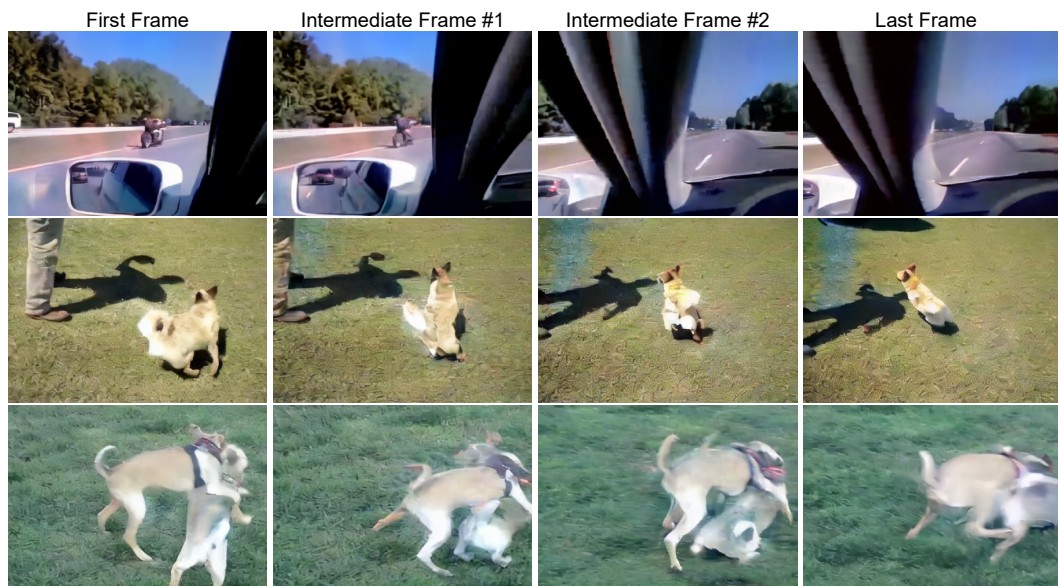

Figure 11: Event-driven interpolation at arbitrary timestamps.

event groups are re-partitioned into 7 new event groups, each covering a new temporal sub-interval between $V_0$ and $V_{10}$. Together with the original event groups ($\mathcal{G}_0$ and $\mathcal{G}_{11}$) aligned with the first and last frames ($V_0$ and $V_{11}$), now we can obtain a total of 9 event groups.

Under this setting, we apply the same interpolation scheme as in the main paper, but now conditioned on the new sequence of 9 event groups. As shown in Figure 11, UniE2F successfully reconstructs the video frame at the *end timestamp* of intermediate event group, even though the temporal duration of this event group has been changed. This demonstrates that, by flexibly adjusting the temporal duration of event group at inference time, our UniE2F can generate frame at arbitrary user-specified timestamp, rather than being restricted to a fixed, pre-defined set of timestamps.

## O    COMPARISON OF PERCEPTUAL QUALITY FOR VIDEO FRAME RECONSTRUCTION

The real-world datasets we use contain relatively few scenes, and the color distribution of HS-ERGB may not fully align with real-world conditions, leading to perceptual deviations from natural scenes. Since UniE2F is trained with RGB frames from real-world scenes in TrackingNet as supervisory targets, the color distribution of the reconstructed frames tends to be closer to real-world distributions, as seen in rows 3 and 4 of Figure 3 of main paper. As a result, perceptual differences in high-dimensional feature spaces, such as those captured by LPIPS, may appear larger. Thus, here, we evaluate the reconstruction performance of our method and the competing approaches using the FID metric (shown in the Table 15). As can be seen, our method outperforms the competitors in terms of FID (Heusel et al., 2017), further supporting its advantages in generating perceptually high-quality reconstructions.

## P    ILLUSTRATION OF EVENT REPRESENTATION

Given the event stream $\{e_i\}_{i=0}^{N_E-1}$ containing $N_E$ events over a duration of $T$ seconds, each event data in $\{e_i\}_{i=0}^{N_E-1}$ is encoded in the format of $e_i = (x_i, y_i, t_i, p_i)$, where $x_i$, $y_i$, $t_i$ and $p_i$ denote the pixel positions, timestamp and the polarity of intensity change. Notably, $x_i \in \{0, ..., W-1\}$, $y_i \in \{0, ..., H-1\}$, $t_i \in [0, T]$, $p_i \in \{+1, -1\}$ for all $i \in \{0, ..., N_E - 1\}$, where $H$ and $W$ denote the height and width of the sensor array of an event camera. Due to the sparse and asynchronous characteristics of the event stream, a typical preprocessing approach in previous work (Rebecq et al., 2019; Stoffregen et al., 2020; Weng et al., 2021) is to accumulate the sequence of event data into

Table 15: Quantitative comparison of perceptual quality using FID (lower is better) for video frame reconstruction. UniE2F achieves the best FID, demonstrating its advantage in generating perceptually natural frames.

| Method | Real-World FID $\downarrow$ | Synthetic FID $\downarrow$ |
|---|---|---|
| E2VID (Rebecq et al., 2019) | 223.7926 | 179.1581 |
| FireNet (Scheerlinck et al., 2020) | 241.5848 | 172.3014 |
| E2VID+ (Stoffregen et al., 2020) | 240.4527 | 203.2372 |
| FireNet+ (Stoffregen et al., 2020) | 250.4045 | 245.4307 |
| ETNet (Weng et al., 2021) | 252.4636 | 207.1250 |
| SSL-E2VID (Paredes-Vallés & De Croon, 2021) | 252.2566 | 241.1340 |
| SPADE-E2VID (Cadena et al., 2021) | 248.8114 | 193.7386 |
| CUBE (Zhao et al., 2024) | 207.4970 | 192.1830 |
| HyperE2VID (Ercan et al., 2024) | 272.1271 | 224.4130 |
| **UniE2F (Ours)** | **184.2509** | **57.1092** |

a 2D image-like tensor representation, allowing its compatibility with frame-based reconstruction algorithms. Specifically, let $\mathbf{V} \in \mathbb{R}^{F \times 3 \times H \times W}$ denote the sequence of $F$ video frames to be reconstructed, and let $\{s_f\}_{f=0}^{F-1}$ be the corresponding timestamps, uniformly distributed over the same duration of $T$ seconds. By setting $s_{-1} = 0$, we distribute the continuous events between two adjacent frames into the group $\mathcal{G}_f = \{e_i \mid \mathbf{s}_{f-1} \leq t_i < \mathbf{s}_f\}$, where $\mathcal{G}_f$ denotes the $f$-th group of events that spans a duration of $\Delta T$ seconds. Then, to leverage the rich generative prior of the pretrained diffusion model developed on 3-channel RGB frames, we transform $F$ event groups into the sequence of 3-channel event representations also with the number of $F$, which is denoted as $\mathbf{E} \in \mathbb{R}^{F \times 3 \times H \times W}$. For each event representation $\mathbf{E}_f$ ($f \in \{0, ..., F-1\}$), we encode the event data of $\mathcal{G}_f$ into three channels corresponding to (i) the sum of all events, (ii) the sum of positive events only, and (iii) the sum of negative events only.

## Q    LIMITATION

In this work, our primary objective is to fully leverage the generative power of diffusion models conditioned on event data, significantly improving the visual quality and temporal consistency of reconstructed frames. Although the proposed UniE2Fhas provides a high-quality baseline for event-to-frame reconstruction, we acknowledge that the use of a pre-trained SVD model's generative prior incurs higher computational cost and GPU memory usage than non-diffusion methods, which may not be suitable for real-time or resource-limited scenarios. In future work, building on the proposed UniE2F, we will investigate diffusion-model distillation, reducing the number of sampling steps, and network pruning to speed up inference for real-time applications.

## R    USE OF LLMs

Large Language Models (LLMs) were used solely to assist with language polishing and minor improvements in writing clarity. All ideas, methodologies, experiments, and analyses presented in this paper are the authors' original work.

## S    VISUAL RESULT

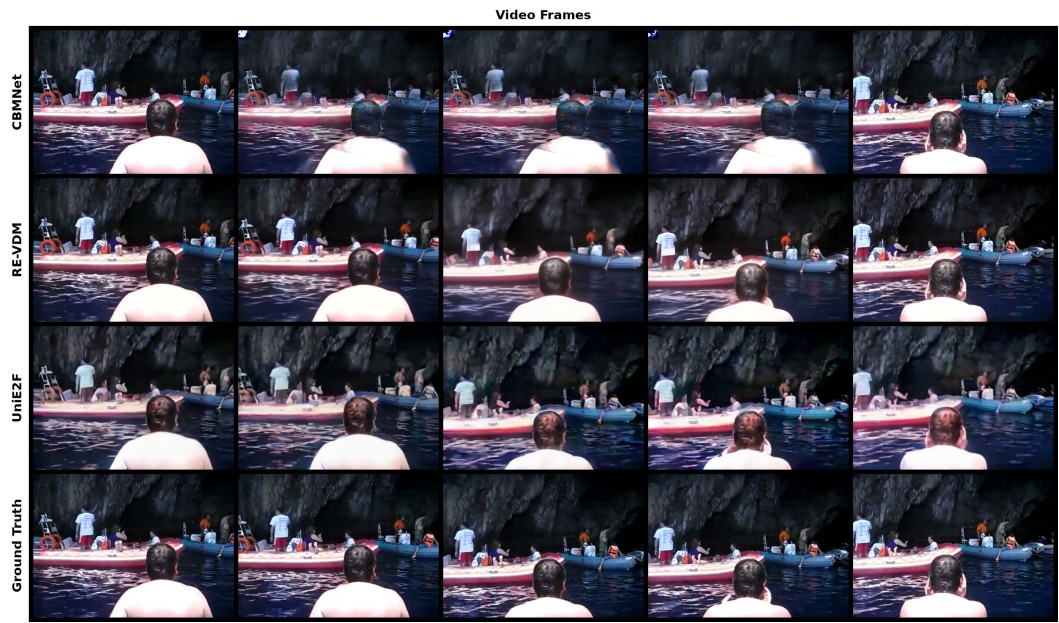

Figure 12: Visual comparison of event-based video frame Interpolation results on synthetic dataset.

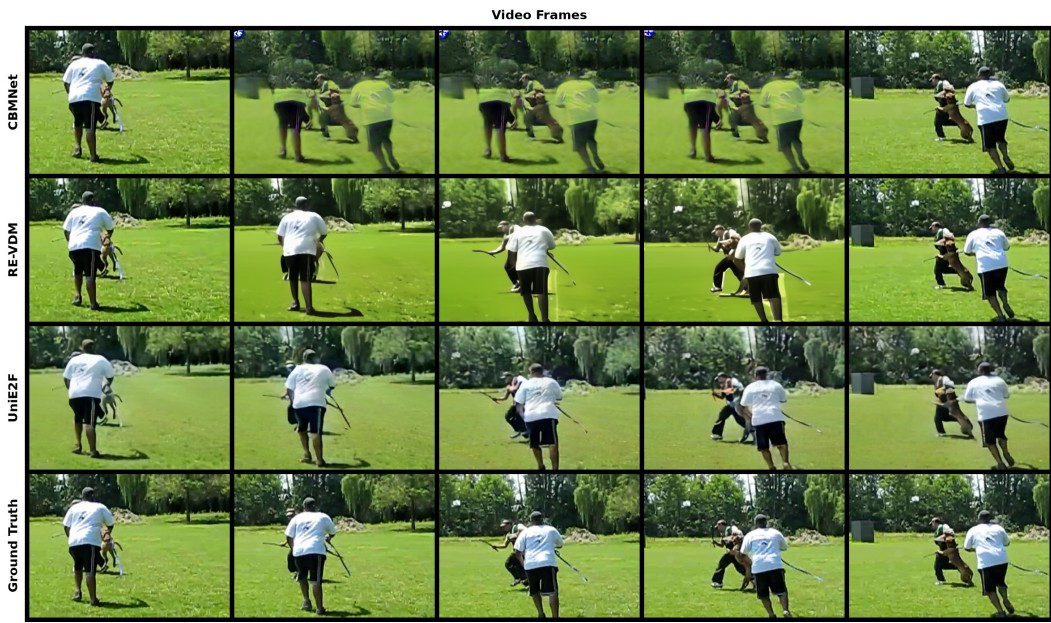

Figure 13: Visual comparison of event-based video frame Interpolation results on synthetic dataset.

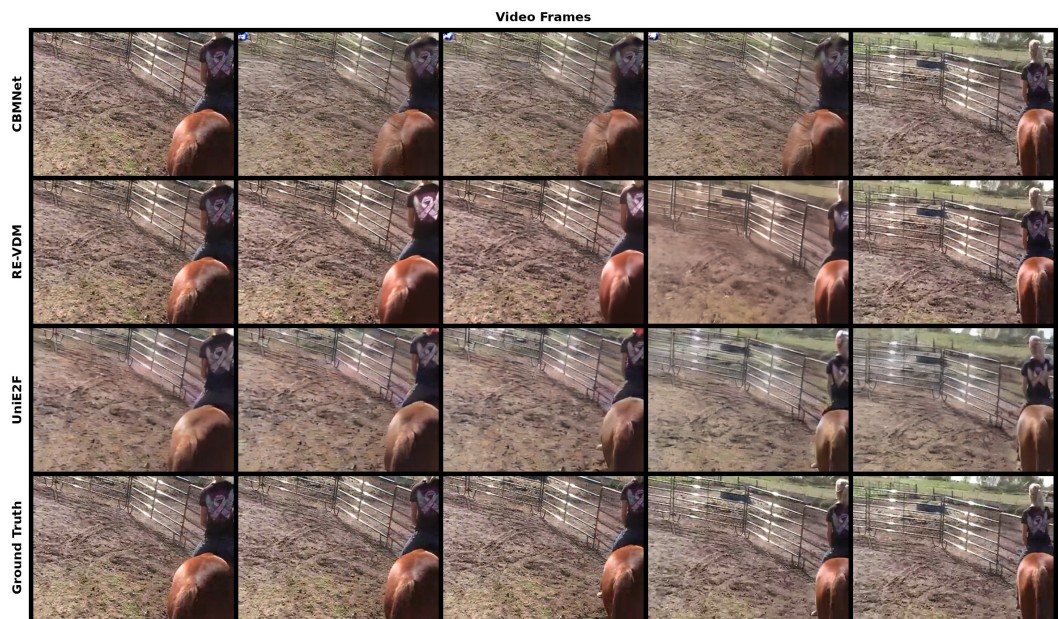

Figure 14: Visual comparison of event-based video frame Interpolation results on synthetic dataset.

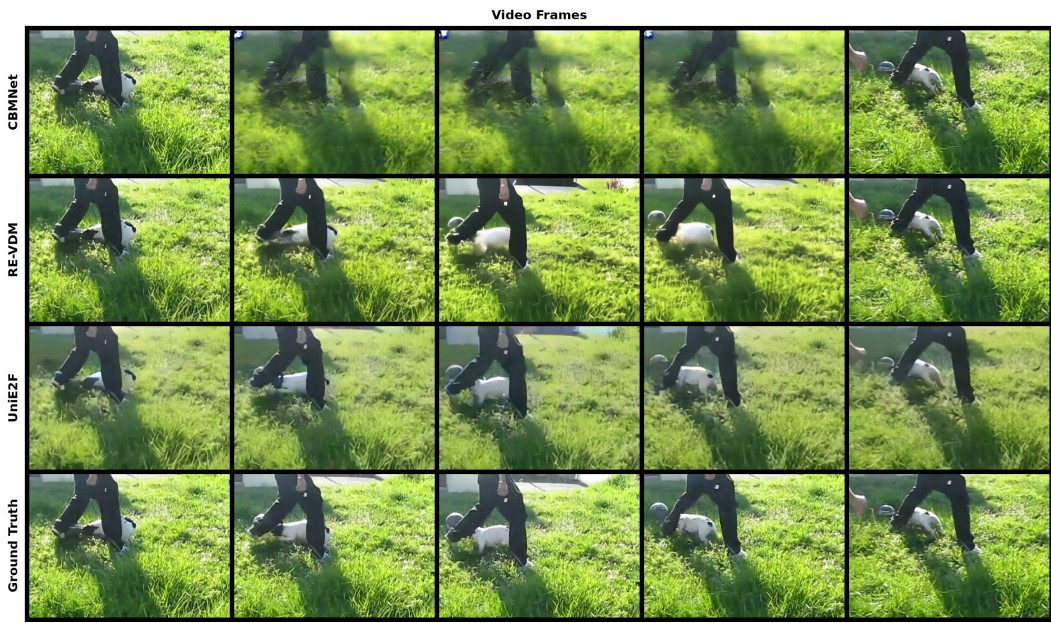

Figure 15: Visual comparison of event-based video frame Interpolation results on synthetic dataset.

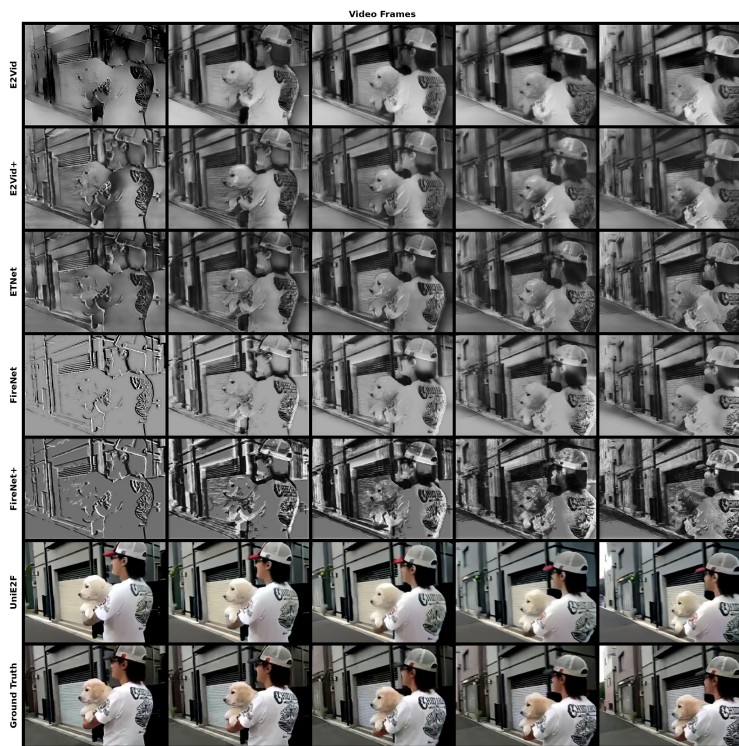

Figure 16: Visual comparison of event-based video frame reconstruction results on synthetic dataset.

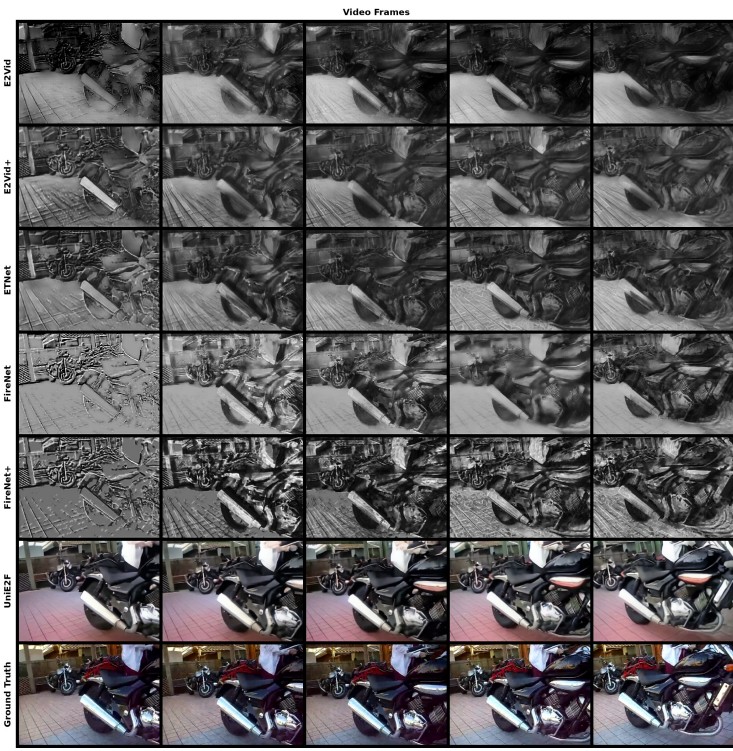

Figure 17: Visual comparison of event-based video frame reconstruction results on synthetic dataset.

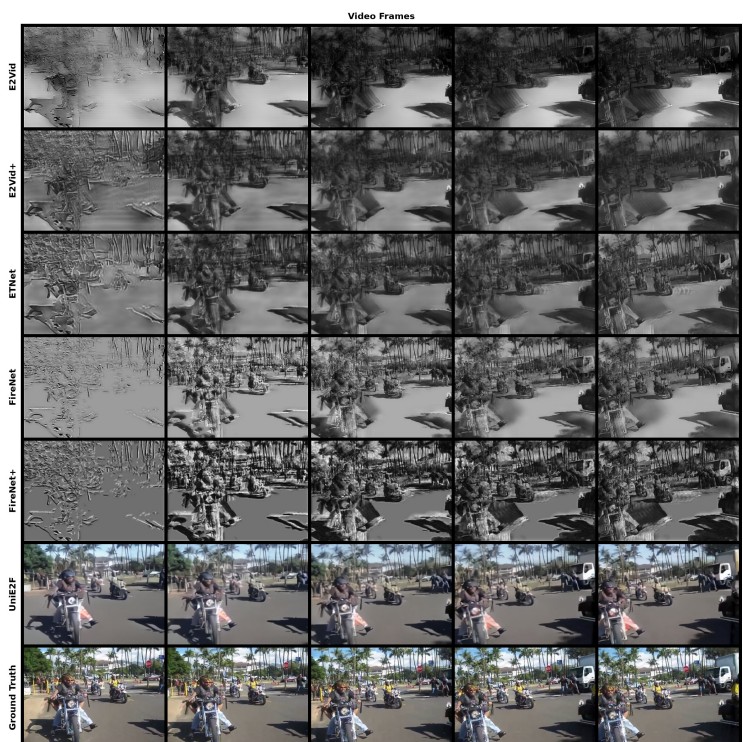

Figure 18: Visual comparison of event-based video frame reconstruction results on synthetic dataset.

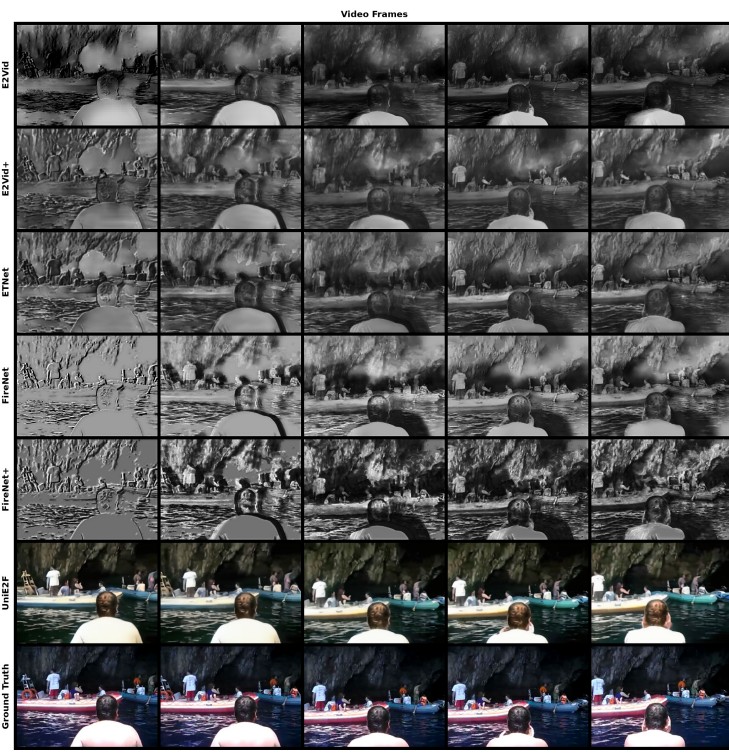

Figure 19: Visual comparison of event-based video frame reconstruction results on synthetic dataset.

