# OpenReview forum: "UniE2F: A Unified Framework for Event-to-Frame Reconstruction with Diffusion Model"
_ICLR.cc/2026/Conference — Submitted to ICLR 2026_

### Official Review · Reviewer_JWWk · 2025-10-22

**Soundness:** 2
**Presentation:** 2
**Contribution:** 2
**Rating:** 2
**Confidence:** 4

**Summary:**

The authors propose UniE2F, a framework for event-to-frame video reconstruction using a fine-tuned Stable Video Diffusion model. While the residual guidance and zero-shot adaptation are positioned as core contributions, they are conceptually weak, insufficiently motivated for real-world scenarios, and lack rigorous empirical validation. The residual guidance builds on known ideas without demonstrating clear necessity or advantage, and the zero-shot claims are not convincingly benchmarked. Overall, the work offers limited novelty and fails to establish meaningful improvements over prior art.

**Strengths:**

The paper introduces a technically sound residual guidance mechanism and a unified zero-shot adaptation strategy, both grounded in prior concepts. While synthetic results are strong and ablations are thorough, the novelty is limited and real-world relevance remains unclear. The appendix and video were informative.

**Weaknesses:**

The paper's introduction and problem setup focus exclusively on reconstruction. It never motivates why the main contributions, i.e., event-based VFI or VFP are important problems to solve or what their real-world applications are.

A key benefit of event cameras is their high temporal resolution, allowing reconstruction at any arbitrary time. The proposed interpolation method (confirmed in App. D) generates a fixed sequence of frames, not a single frame at an arbitrary timestamp t. This misses the primary advantage of the sensor.

One significant weakness for applicability is the prohibitive computational cost. Appendix F (Table 8) reveals the method requires 46,753 MB (~47 GB) of VRAM and 245.364 TMACs, which is orders of magnitude more than competitors (e.g., HyperE2VID at 1052 MB and 0.060 TMACs). The 48-second latency for 12 frames (Sec 5.1) makes this method completely unusable for any practical or real-time application, which is the entire point of event cameras.

The paper claims SOTA on real-world reconstruction, but this is only true for MSE/SSIM. The LPIPS score (Table 1) is significantly worse than prior work (0.674 vs 0.562). This major discrepancy, which suggests poor perceptual quality, is never discussed or explained.

The key "zero-shot" contribution (VFI/VFP) does not generalize well. Table 2 shows it is clearly outperformed by standard baselines (e.g., CBMNet) when they are simply retrained on the target data. This suggests the "unification" is more of a curiosity on synthetic data than a robust, general-purpose tool.

**Questions:**

Major:

- The LPIPS scores in Table 1 (Reconstruction) and Table 2 (VFI/VFP) are consistently and significantly worse than baselines on real-world data. This contradicts the excellent qualitative results in Figure 3. Can you explain this discrepancy? Is the LPIPS metric failing, or are the qualitative examples cherry-picked? This is a crucial point of confusion.

- The computational costs (Table 8, ~47GB VRAM) and latency (48s) are prohibitive and render the method unusable for any practical event-camera application. The limitation section (App. G) mentions this, but I'd like to ask: Do you believe this is a fundamental limitation of using large diffusion models for this task, or do you have concrete evidence that distillation/pruning can bridge the orders-of-magnitude gap to competitors?

- Your interpolation method (Sec 4.3, App. D) generates a fixed sequence of frames (e.g., 10 frames between $V_0$ and $V_{11}$). Why did you not pursue a more "event-native" approach, such as reconstructing a single frame at an arbitrary timestamp $t \in (0, 11)$?

- Could you expand on the motivation and real-world applications for zero-shot event-based interpolation and prediction? The paper currently justifies reconstruction but not these other tasks.

Minor:
- The introduction, preliminary, and related work sections are quite lengthy, spanning nearly four pages, yet they include redundant background and omit discussion of several key prior works. Could the authors clarify their criteria for selecting related work, particularly regarding early image reconstruction methods and event stacking approaches that are not cited?

Suggestions:
Figure 1 illustrates general diffusion model concepts but does not appear to convey any paper-specific insights.
There are more important tables from the appendix that can be moved to the main paper if such sections become shorter.

---

> ### Author Response · Authors · 2025-11-25
> **Official Comment by Authors**
>
> **Q: Motivation and Applications of Event-based VFI and VFP**
>
> **Response**: We agree that the current introduction allocates more space to the reconstruction problem itself, while the motivation and practical value of event-based VFI and VFP are not sufficiently emphasized, and **we have strengthen this aspect in the introduction section of the resubmitted manuscript**. Besides, in "Related Work", we already discussed the role of event-based VFI and summarized existing studies. Recent works on event-based VFI/VFP have attracted growing attention precisely because event-driven interpolation and prediction are important in practical applications: for example, in smartphone cameras, event data can support **high-speed, high-dynamic-range video capture and frame interpolation to obtain smoother slow-motion effects**; in scientific imaging and high-speed experimental observation, event data can similarly compensate for the limitations of conventional frame-based cameras under **extremely fast motion and low-light conditions**. **We have added a more explicit task definition and application discussion for event-based VFI/VFP to the introduction in the revised manuscript.**
>
> **Q: Computational Cost, Performance, and Practical Applicability**
>
> **Response**: We want to note that our work is designed to **leverage the great potential to explore a effective event-based frame reconstruction algorithm**, which can really reconstruct the real-world sequence, instead of toy experiments in the limited dataset.  Moreover, it is worth emphasizing that we do not regard **“practical”** as synonymous with **“real-time”**. **UniE2F can be practical for offline video reconstruction**, where many practical applications can tolerate second-level or even longer latency, such as slightly delayed quality enhancement of videos captured on mobile devices in the cloud or on device,  multi-exposure HDR reconstruction, or scientific purpose high-speed video capturing. In these scenarios, reconstruction **quality and robustness** are usually more critical than real-time response. Although UniE2F incurs higher computational overhead due to the use of an SVD-pre-trained model, the results in Sec. 5.2 show that our method achieves **substantially better** reconstruction quality than existing non-diffusion-based approaches, **and produces results that are closer to real scenes**.  Moreover, even compared with CUBE (Zhao et al., 2024), another diffusion-based event-driven frame reconstruction method, our approach requires less computation while achieving better quantitative metrics and visual quality. Finally, since the current implementation does not employ any model compression or acceleration techniques, **the memory footprint and computational cost reported in the Appendix can be viewed as an upper bound without engineering optimization**. In future work, we will explore network pruning, distillation, and consistency-model-based acceleration to reduce the number of sampling steps and computational cost while preserving reconstruction quality, thereby improving the applicability of UniE2F in resource-constrained scenarios.
>
>
>
> **Q: Clarification on Zero-Shot VFI/VFP Generalization**
>
> **Response**: In response to the reviewer’s concern regarding the zero-shot generalization of VFI/VFP, we would like to clarify further. On real-world datasets, UniE2F is applied to VFI/VFP tasks in a completely zero-shot setting, whereas comparison methods (e.g., CBMNet) are fine-tuned for interpolation. In this setup, compared to retrained CBMNet, UniE2F performs slightly worse in VFI-4×, **but in the long-range VFI-11× task, it achieves better MSE and comparable LPIPS**. It is important to note that **UniE2F was not specifically trained for interpolation or prediction: no explicit learning of optical flow, reference frame warping, or feature-level aggregation was performed**. Instead, UniE2F utilizes a unified event-to-frame reconstruction framework, adjusting the reverse diffusion process to achieve VFI/VFP. This shows that, without warping or motion estimation, UniE2F still performs on par with, or even better than, baselines designed and trained for interpolation and prediction tasks. **This highlights the practical applicability and generalization potential of our proposed zero-shot framework for event-driven VFI/VFP tasks**, beyond just synthetic data demonstrations.

---

> ### Author Response · Authors · 2025-11-25
> **Official Comment by Authors**
>
> **Q: Comparison of Perceptual Quality**
>
> **Response**: In terms of the LPIPS score (0.674 vs. 0.562), we would like to provide some context for this difference. The real-world datasets we used contain **relatively few scenes**, and **the color distribution of HS-ERGB may not fully align with real-world conditions, leading to perceptual deviations from natural scenes**. Since UniE2F was trained with RGB frames from real-world scenes in TrackingNet as supervisory targets, the color distribution of the reconstructed frames tends to be closer to real-world distributions, as seen in rows 3 and 4 of Figure 3 of main paper. **As a result, perceptual differences in high-dimensional feature spaces, such as those captured by LPIPS, may appear larger**. Furthermore, we also evaluated the reconstruction performance of our method and the competing approaches using the FID metric (shown in the table below). As can be seen, **our method outperforms the competitors in terms of FID, further supporting its advantages in generating perceptually high-quality reconstructions**.
>
>
> [Synthetic Dataset]
> | Method      | FID ↓        |
> |:------------|:-------------|
> | E2VID       | 179.1581 |
> | FireNet     | 172.3014 |
> | E2VID+      | 203.2372|
> | FireNet+    | 245.4307 |
> | ETNet       | 207.1250 |
> | SSL-E2VID   | 241.1340  |
> | SPADE-E2VID | 193.7386 |
> | CUBE        | 192.1830 |
> | HyperE2VID  | 224.4130 |
> | UniE2F      | **57.1092**|
>
> [HS-ERGB Dataset]
> | Method      | FID ↓        |
> |:------------|:-------------|
> | E2VID       | 223.7926     |
> | FireNet     | 241.5848     |
> | E2VID+      | 240.4527     |
> | FireNet+    | 250.4045     |
> | ETNet       | 252.4636     |
> | SSL-E2VID   | 252.2566     |
> | SPADE-E2VID | 248.8114     |
> | CUBE        | 207.4970     |
> | HyperE2VID  | 272.1271     |
> | UniE2F      | **184.2509** |
>
>
> **Q: Computational Cost, Latency, and Real-Time Deployment**
>
> **Response**: Regarding computational cost and latency, our proposed UniE2F is **primarily designed for offline or cloud-based reconstruction scenarios, aiming to fully exploit the generative capability of pre-trained diffusion models on event data rather than targeting real-time deployment under extremely resource-constrained scenarios**. Similar to many large-scale models that are naturally deployed in the cloud instead of on-device, not every network is required to run locally; in fact, most high-performance models today cannot operate at real-time rates on microcontrollers or other edge platforms. Additionally, **we do not consider computational overhead to be a fundamental limitation to applying large-scale diffusion models in vision tasks that involve event cameras**. The computational cost of diffusion models is mainly determined by model size and the number of sampling steps, and prior works have shown that techniques such as model distillation, network pruning, and consistency models can reduce the sampling steps to as few as 1–2 while significantly reducing the parameter count and memory footprint, thereby achieving orders-of-magnitude acceleration without sacrificing reconstruction quality.
>
> **Q: Motivation and Real-World Applications of Zero-Shot Event-Based Interpolation and Prediction**
>
> **Response**: Our motivation is as follows: **on the relatively more challenging reconstruction task, we first inject rich event-to-frame knowledge into the network through pre-training, obtaining a general event-based frame processing pre-trained model**. On this basis, we reformulate the reverse diffusion sampling process so that, even without task-specific training for interpolation and prediction, the model can naturally generalize to these tasks in a zero-shot manner. This enables a single model in practical application to jointly support event-based video frame reconstruction, interpolation, and prediction, **significantly reducing the cost of separately training and deploying models for different tasks**. The experimental results also show that UniE2F achieves zero-shot performance on interpolation and prediction comparable to specially trained methods on real datasets. **This confirms its generic event-to-frame domain transfer capability and lays the foundation for extending to more  event-to-frame tasks**. Moreover, it is analogous to pre-trained large language models that generalize across multiple NLP tasks—just as pre-trained LLMs can achieve further gains after fine-tuning on downstream tasks, **UniE2F is also expected to benefit from additional fine-tuning on dedicated interpolation/prediction data**.

---

> ### Author Response · Authors · 2025-11-25
> **Official Comment by Authors**
>
> **Q: Clarification on the Selection of Related Work**
>
> **Response**: Our selection of related work focuses on methods **that are most directly comparable to our approach and representative in event-to-frame reconstruction, interpolation, and prediction**. This includes early reconstruction methods based on event intensity gradients as well as more recent approaches using convolutional networks and diffusion models. Regarding event stacking approaches, **we have additionally included in the related work section citations** to event stacking methods.
>
> **Q: Event-Based Frame Interpolation for Arbitrary Timestamp**
>
> **Response**: In our UniE2F method, we can generate frames at arbitrary timestamps by **adjusting the temporal duration of each event group ($\Delta_T$) and iteratively performing interpolation**. Specifically, by inserting 9 frames between two given frames, we can obtain frames at timestamps of {0.1, 0.2, ..., 0.9}. For finer-grained frames at more specific timestamps, **we can iteratively perform additional interpolation between the adjacent frames from the previous interpolation round**, progressively narrowing down the temporal duration $\Delta_T$ to generate frames at arbitrary timestamp.
>
> **Q: Illustration of Figure 1**
>
> **Response**: Figure 1 not only illustrates a generic diffusion process; instead, it focuses on **characterizing how UniE2F behaves on three event-based tasks**. On the left, we show the input configurations for the three tasks, respectively: events only (reconstruction), events + first frame (prediction), and events + first and last frames (interpolation). On the right, panels A–C show the reconstructed results for the three tasks, while panel D shows the corresponding ground-truth frame sequences. **In the middle, each pair of solid and dashed curves with the same color denotes the reverse sampling trajectories of an SDE and an ODE under the corresponding task setting.** They start from the same noise state and evolve toward the target frames shown in A/B/C, providing an intuitive visualization of how different reverse sampling processes lead to different generative trajectories.
>
> **Q: Organization of the paper contents**
>
> **Response**: Due to length limitations, we initially placed these tables in the Appendix; however, we agree that some of them are essential for the main paper. In the revised version, **we have shortened the corresponding sections and move the most informative tables** from the Appendix into the main paper.
>
> **NOTE: We are currently conducting experiments on frame interpolation at arbitrary timestamps and will report the results as soon as they are available.**

---

> ### Comment · Reviewer_JWWk · 2025-11-26
> **Acknowledging the rebuttal**
>
> Thank you for the thoughtful rebuttal, additional metrics (such as the FID tables), ablations, and updates to the manuscript. These additions provide helpful context and demonstrate effort in addressing reviewer feedback.
>
> That said, several key concerns remain unresolved, aligning with points raised by my fellow reviewers (e.g., computational inefficiency noted by FKni, V449, and Bq1Z; perceptual discrepancies and real-world generalization highlighted by V449 and Bq1Z; and methodological details emphasized by FKni).
>
> Specifically:
> - Arbitrary timestamp interpolation: The iterative approach is a reasonable suggestion, but without demonstrated results, it leaves practicality unclear, missing a core event camera advantage.
> - Computational costs and latency: While future acceleration is promising, no concrete evidence closes the substantial gap to baselines, echoing the efficiency critiques from other reviewers.
> - LPIPS discrepancies and perceptual quality: The FID scores are a positive step, but the explanation for LPIPS on real data requires further validation to rule out issues like cherry-picking, consistent with Bq1Z's and V449's observations on real-world performance.
> - Zero-shot generalization and motivation: The expanded applications are appreciated, but performance still trails retrained baselines on real data, limiting the unification's robustness, as also implied in V449's fairness concerns.
>
> Given these persistent issues, shared across the reviews, I believe the contribution falls short for ICLR acceptance. Rating unchanged.

---

> ### Author Response · Authors · 2025-11-26
> **Response to Reviewer JWWk**
>
> Hi, Reviewer **JWWk**,
>
> We wish to express our concern that the review does not reflect the responsibilities expected from a proper academic perspective. **Such an approach risks undermining the integrity of the community**.
>
> We believe that we have clearly addressed the issues raised by you. Moreover, high computational cost is a very common challenge for modern large-scale models. **Treating current runtime/memory as a decisive reason for rejection is not fully aligned with the prevailing direction of foundation-model-based research**, where adapting large pretrained models has become a central theme. Besides, regarding the computational cost of our method, we have clarified that **there already exists a substantial body of work** on accelerating diffusion models. For the **LPIPS concern**, we have explained in our response “Comparison of Perceptual Quality” that **this is due to the inherent limitation of real-world data**. Finally, we emphasize that a key strength of our approach is its ability to achieve competitive zero-shot performance on real-world data, even when compared with baselines that are retrained on the target datasets. **In particular, we have explicitly shown that, on the long-range VFI-11× setting, our zero-shot UniE2F attains performance comparable to CBMNet, despite CBMNet being specifically retrained in the interpolation setting**.
>
> We are concerned that you appears to be largely unfamiliar with this research area and therefore is unable to provide constructive suggestions to improve the quality of our paper. **Consequently, we do not believe you are fulfilling the responsibilities from a proper academic review perspective. With this irresponsible attitude, this community will only continue to get worse.** And we will not provide further responses to the your remaining questions.
>
> Authors

---

> > ### Comment · Reviewer_JWWk · 2025-11-26
> > **Acknowledging Author's Response to Reviewer**
> >
> > Dear Authors,
> >
> > Thank you for your response.
> >
> > I stand by my original technical assessment and the concerns I raised (computational feasibility, lack of true arbitrary-timestamp reconstruction, unexplained perceptual metric discrepancies, and limited zero-shot generalization on real data).
> > These points were shared, in whole or in part, by the three other reviewers.
> >
> > The rebuttal did not provide the requested concrete evidence or experiments on the most critical issues, and the newly added personal remarks about my familiarity with the field or my responsibilities as a reviewer are both inaccurate and irrelevant to the scientific discussion.
> >
> > No further response from me is needed.
> > I leave the finalize my score as originally submitted.

---

> ### Author Response · Authors · 2025-12-03
> **Official Comment by Authors**
>
> **Q: Event-Driven Interpolation at Arbitrary Timestamps**
>
> **R:** Here, we provide the experiment to evaluate the interpolation capability of our UniE2F at arbitrary timestamps. Concretely, we take a sequence of 12 frames. Concretely, we take a sequence of 12 frames $\{ V_0, V_1, \dots, V_{11} \}$ from the dataset and the 12 corresponding event groups $\{ G_{0}, G_{1}, \dots, G_{11} \}$. Then, for the interpolation task, we **focus only on the frames between the first and last frames, i.e., the 10 event groups $\{G_{1}, G_{2}, \dots, G_{10} \}$ associated with $\{V_1, V_2, \dots, V_{10} \}$**. These 10 intermediate event groups are **re-partitioned** into 7 new event groups, each covering a new temporal sub-interval between $V_0$ and $V_{10}$. Together with the original event groups ($G_{0}$ and $G_{11}$) aligned with the first and last frames ($V_0$ and $V_{11}$), now we can obtain a total of 9 event groups.
>
> Under this setting, we apply the same interpolation scheme as in the main paper, but now conditioned on the new sequence of 9 event groups. As shown in Figure 11 of the Appendix, **UniE2F successfully reconstructs the video frame at the end timestamp of intermediate event group, even though the temporal duration of this event group has been changed**. This demonstrates that, by **flexibly adjusting the temporal duration of event group** at inference time, our UniE2F can generate frame at **arbitrary user-specified timestamp**, rather than being restricted to a fixed, pre-defined set of timestamps.

---

### Official Review · Reviewer_Bq1Z · 2025-10-26

**Soundness:** 2
**Presentation:** 3
**Contribution:** 2
**Rating:** 6
**Confidence:** 4

**Summary:**

This paper proposes a unified framework UniE2F for event-to-frame reconstruction. By fusing the generative prior of pre-trained video diffusion models with the physical characteristics of event data, it enhances the quality of reconstructing high-fidelity video frames from sparse events, and its effectiveness has been verified through experiments. The main contributions and ideas of the paper can be summarized as follows:

1. Unified Task Framework
(1) Event-driven Frame Reconstruction: Based on the pre-trained Stable Video Diffusion (SVD) model, event data is encoded into 3-channel tensors as conditional inputs. The model learns the mapping relationship between events and video frames through fine-tuning, establishing fundamental reconstruction capabilities.
(2) Inter-frame Residual Guidance Mechanism: Leveraging the physical correlation between events and inter-frame brightness changes, a ResNet-based inter-frame residual prediction module is introduced. The latent variables of the diffusion model are optimized via gradient descent to improve the temporal consistency and accuracy of reconstructed frames.
(3) Zero-shot Interpolation and Prediction: By modulating the score function of the reverse diffusion process, the prior knowledge from the reconstruction task is transferred to video frame interpolation and prediction tasks without additional training, enabling unified handling of "reconstruction-interpolation-prediction".

2. Core Technical Innovations An event-based inter-frame residual guidance strategy is proposed. Theoretical proof demonstrates that its gradient aligns with the tangent space of the data manifold learned by the diffusion model, ensuring the optimization process does not compromise generation quality. A score function modulation method is designed, which uses the latent variable deviation of reference frames to correct the sampling process, enhancing the temporal consistency and visual fidelity of interpolation/prediction tasks.

3. Data Construction and Experimental Validation Training data is synthesized from real-world videos, and tests are conducted on both real-world and synthetic datasets. Extensive experiments and ablation studies verify that UniE2F outperforms most existing state-of-the-art (SOTA) models.

**Strengths:**

1. This paper proposes a systematic three-stage event-to-frame reconstruction framework: Event-Based Video Frame Reconstruction → Inter-Frame Residual Prediction Training → Inter-Frame Temporal Residual Guidance. Ablation studies validate the effectiveness of each component. Innovatively integrating the generative prior of the pre-trained Stable Video Diffusion (SVD) model with the physical characteristics of event data, the method achieves unified handling of "video frame reconstruction-interpolation-prediction" through three core modules: event-conditioned fine-tuning, inter-frame residual guidance, and score function modulation. It breaks through the limitation of traditional methods confined to single tasks and can adapt to interpolation and prediction tasks in a zero-shot manner without additional training.

2. An event-based inter-frame residual guidance mechanism is designed. It predicts inter-frame residuals via ResNet and optimizes the latent variables of the diffusion model combined with gradient descent. Meanwhile, it is theoretically proven that this gradient aligns with the tangent space of the data manifold, ensuring the optimization does not compromise generation quality and the reconstruction error is bounded. This effectively enhances the temporal consistency and detail accuracy of the reconstructed frames.

3. Relevant experiments verify the effectiveness of the proposed modules. Compared with other methods, UniE2F achieves state-of-the-art (SOTA) performance.

4. The paper is logically structured and easy to understand.

**Weaknesses:**

1. Testing is only conducted on sequences extracted from TrackingNet and HS-ERGB, without validation on other datasets, making it impossible to demonstrate the true effectiveness and generalization of the method.

2. Specific details of the used Stable Video Diffusion (SVD) are not provided, such as the pre-trained model employed, parameter count, and other relevant specifications.

3. The originality is insufficient. From a methodological perspective, introducing residual guidance optimization is one of the core innovations of the paper, but the ablation experiments show that its improvement on performance is not significant.

4. Figure 2 (the method framework) could be more detailed. For example, relevant components of the training phase should be added.

5. Although the method achieves state-of-the-art (SOTA) performance on the synthetic test set, it exhibits suboptimal performance in most cases on real-world datasets.

6. Compared with suboptimal methods, while this method achieves certain performance improvements, its computational cost and memory footprint are several times or even dozens of times higher than those of traditional methods.

**Questions:**

1. The paper only verifies performance on the TrackingNet and HS-ERGB datasets, without involving datasets for extreme scenarios such as low light and fast motion. How robust is the method in such complex scenarios? Will performance degrade due to a sudden increase in event sparsity or noise interference?

2. Regarding the used SVD model, the network structure details and parameter count are not specified. Could you provide supplementary explanations? Have you tried SVD models with different parameter counts?

3. Could the method be trained on real-world datasets and verified for effectiveness on real-world datasets?

4. Is the way of encoding event data into 3-channel tensors (sum of all events, sum of positive events only, sum of negative events only) the optimal choice? Have you tried temporal dimension encoding (e.g., event occurrence frequency)?

---

> ### Author Response · Authors · 2025-11-25
> **Official Comment by Authors**
>
> **Q: Detailed Specifications of Stable Video Diffusion**
>
> **Response**: Stable Video Diffusion uses the U-Net of Stable Diffusion 2.1 as its backbone, extending the original 2D image diffusion network into a spatial-temporal U-Net with an temporal dimension. Specifically,  temporal convolution layers and temporal attention layers are inserted between the U-Net’s residual blocks and self-attention blocks, enabling features to propagate and interact across frames and thus form a coherent spatial-temporal representation. The entire model performs the diffusion process on video frames in the latent space. At sampling time it is conditioned on the first frame, text description, and frame rate to control both the semantic content and temporal dynamics of the generated video. **In addition, the pre-trained model contains 1,521M parameters**.
>
> **Q: Effectiveness of Inter-Frame Residual Guidance**
>
> **Response**: The Inter-Frame Residual Guidance (IFRG) plays a crucial role in improving the accuracy of frame reconstruction by providing additional constraints between adjacent frames, helping the model better capture temporal coherence. To demonstrate its effectiveness, we have included a visual comparison of the variants in Appendix. **As shown in the Figure 7, without IFRG (top row), the reconstructions exhibit noticeable blurring and structural distortion: the bars of the fence are wavy and over-smoothed, and the ground textures are largely washed out. With IFRG (middle row), edges and fine structures align much better with the ground truth (bottom row): the fence bars are straighter and more regular, the sand and background textures are more faithfully recovered.** These improvements indicate that IFRG effectively constrains the reconstruction with inter-frame intensity changes, yielding frames that are both structurally more accurate to the ground truth.
>
> **Q: Detailed Illustration of the Framework Structure**
>
> **Response**: We have created a more detailed Figure 6 illustrating the structure of our UniE2F, including relevant details of the training phase, and have included it in the Appendix for further clarity.
>
> **Q: Real-World Performance and Perceptual Quality**
>
> **Response**: We respectfully disagree that our method exhibits suboptimal performance on real-world datasets in most cases. As shown in Table 1 of main paper, **our method achieves the best MSE and SSIM on the real-world datasets**. In terms of the LPIPS score, we would like to provide some context for this difference. The real-world datasets we used contain relatively few scenes, and **the color distribution of HS-ERGB may not fully align with real-world conditions**, leading to perceptual deviations from natural scenes. Since UniE2F was trained with RGB frames from real-world scenes in TrackingNet as supervisory targets, the color distribution of the reconstructed frames tends to be closer to real-world distributions, as seen in rows 3 and 4 of Figure 3 of main paper. **As a result, perceptual differences in high-dimensional feature spaces, such as those captured by LPIPS, may appear larger.**
>
> In addition, we computed the Fréchet Inception Distance (FID) for both the previous methods and ours on the synthetic and real-world datasets, with the results summarized in the table below. **As can be seen from the table, our method achieves the best FID scores, which further demonstrates its advantages in terms of overall perceptual quality and distributional consistency.**
>
> [Synthetic Dataset]
> | Method      | FID ↓        |
> |:------------|:-------------|
> | E2VID       | 179.1581 |
> | FireNet     | 172.3014 |
> | E2VID+      | 203.2372|
> | FireNet+    | 245.4307 |
> | ETNet       | 207.1250 |
> | SSL-E2VID   | 241.1340  |
> | SPADE-E2VID | 193.7386 |
> | CUBE        | 192.1830 |
> | HyperE2VID  | 224.4130 |
> | UniE2F      | **57.1092**|
>
> [HS-ERGB Dataset]
> | Method      | FID ↓        |
> |:------------|:-------------|
> | E2VID       | 223.7926     |
> | FireNet     | 241.5848     |
> | E2VID+      | 240.4527     |
> | FireNet+    | 250.4045     |
> | ETNet       | 252.4636     |
> | SSL-E2VID   | 252.2566     |
> | SPADE-E2VID | 248.8114     |
> | CUBE        | 207.4970     |
> | HyperE2VID  | 272.1271     |
> | UniE2F      | **184.2509** |

---

> ### Author Response · Authors · 2025-11-25
> **Official Comment by Authors**
>
> **Q: Computational Overhead**
>
> **Response**: Although our UniE2F incurs higher computational cost due to the incorporation of the pre-trained SVD model, it achieves a **significant improvement** in reconstruction quality over previous methods: the recovered frames exhibit much sharper details, more faithful structures, and more accurate colors, thus being closer to real-world scenes and without the limitation to the specific dataset. In contrast, prior approaches typically produce frames with blurry contours, missing textures, or noticeable color distortions and only work on the data similar to the training set.
>
> We fully agree that accelerating diffusion models is an important and, as shown by existing work, feasible research direction. However, this paper mainly focuses on how to effectively transfer the generative prior to the event-to-frame reconstruction task, in order to push event-to-frame reconstruction quality closer to realistic scenarios. **Addressing both reconstruction-quality improvement and diffusion-model acceleration within a single paper would significantly increase the workload and length, and would also obscure the core contribution.** Therefore, in the current work we prioritize developing and validating the transfer of the generative prior for improving reconstruction quality, and will further explore diffusion model acceleration strategies specifically for the event-to-frame task in future work.
>
> **Q: Robustness to Event Noise and Sparse-Event Failure Cases**
>
> **Response**:In the original version of the paper, **the robustness evaluation of our model under different levels of event noise was provided in Appendix Sec. E “Additional Ablation Study”**. In the updated version, these results have been moved to Sec. 5.4 of the main paper. The results show that **our method maintains strong robustness to event noise across these settings**. Regarding failure cases, we acknowledge that when the event stream becomes extremely sparse, with only a few isolated pixels undergo intensity changes, our method cannot reliably reconstruct fine-grained structures and details. **It is worth noting that this limitation is common to all methods that rely solely on events for video reconstruction**, as the input event signal becomes insufficient. We have included representative qualitative examples of such sparse-event failure cases in Figure 9 of the Appendix.
>
> [Synthetic Dataset]
> | Noise-level Coefficient | MSE ↓      | SSIM ↑     | LPIPS ↓    |
> |:------------------------|:-----------|:-----------|:-----------|
> | 1.0 | 0.0193     | 0.6690     | 0.4280 |
> | 0.5 | 0.0177     | 0.6930     | 0.4050 |
> | 0.1  | 0.0168     | 0.7080     | 0.3950 |
> | Baseline  | **0.0167** | **0.7100** | **0.3940** |
>
> **Q: Architecture and Parameter Count of the SVD Backbone**
>
> **Response**: **Stable Video Diffusion (SVD), which has 1,521M parameters**, uses the U-Net of Stable Diffusion 2.1 as its backbone, extending the original 2D image diffusion network into a spatio-temporal U-Net with a temporal dimension. Specifically,  temporal convolution layers and temporal attention layers are inserted between the U-Net’s residual blocks and self-attention blocks, enabling features to propagate and interact across frames and thus form a coherent spatio-temporal representation. The entire model performs the diffusion process on video frames in the latent space. At sampling time it is conditioned on the first frame, text description, and frame rate to control both the semantic content and temporal dynamics of the generated video. **Since the authors of SVD only release a single pre-trained model with 1,521M parameters, changing the model size prevents direct reuse of its powerful generative prior. In this case, retraining a large-scale video diffusion model from scratch would require a large amount of data and computational cost**, which is unaffordable for academic researchers.
>
> **Q: Rationale for the Three-Channel Event Representation**
>
> **Response**: We agree that the current choice of a 3-channel event encoding is not necessarily the optimal design. We adopt this scheme primarily to maintain structural compatibility with SVD model pre-trained on 3-channel RGB data, **which allows us to directly leverage their strong generative priors without modifying the backbone architecture**. **When compressing continuous and sparse event streams into finite-channel tensors, some temporal information is inevitably lost regardless of the specific encoding strategy**. Therefore, in this work, **we focus on how to best exploit large-scale pre-trained diffusion models for the event-to-frame reconstruction task given the compressed representation, rather than exploring all possible event encoding strategies**. Regarding the reviewer’s suggestion on explicit temporal-dimension encoding, we consider it a valuable direction and plan to investigate such encoding in future work, given current limitations in length of manuscript and computational resources.

---

> ### Author Response · Authors · 2025-11-25
> **Official Comment by Authors**
>
> **NOTE: Due to time constraints, we are still running experiments on additional real-world event datasets and retraining UniE2F conditioned on the real-world dataset, which is highly time-consuming. Consequently, we are not yet able to report these results, but we plan to submit the corresponding performance numbers later this week.**

---

> ### Author Response · Authors · 2025-12-03
> **Official Comment by Authors**
>
> **Q: Comparison on Real-World Datasets (HQF, IJRR, and MVSEC)**
>
> **R:** We compare UniE2F with existing methods on three real-world event camera datasets: HQF, IJRR, and MVSEC, which provide **single-channel intensity frames** as ground truth. As shown in Figure 10 of the Appendix, **UniE2F can reconstruct frames with more realistic colors and clear details, making them closer to real scenes**. In contrast, **the other methods can only produce single-channel grayscale reconstructions, which lack color information and look less natural**.
>
> **Q: Training UniE2F on Real-World Event Data**
>
> **R:** Here, we split the HS-ERGB real-world dataset into a training set and a test set, and trained UniE2F directly on its training split. The quantitative results on the test split are reported in table below. From the table, we observe that **the limited size and diversity of HS-ERGB cause the model to overfit a few specific scenes** when learning the event-to-frame mapping, which in turn **weakens the generalization ability brought by large-scale pre-training**. Therefore, in the main paper, we choose to train UniE2F on a more diverse synthetic dataset and evaluate it on real-world datasets.
>
> | Training Data | MSE ↓   | SSIM ↑  | LPIPS ↓ |
> |--------------|---------|---------|---------|
> | Real-World   | 0.1084  | 0.3080  | 0.7580  |
> | Synthetic    | **0.0167** | **0.7100** | **0.3940** |

---

### Official Review · Reviewer_V449 · 2025-10-27

**Soundness:** 3
**Presentation:** 3
**Contribution:** 2
**Rating:** 4
**Confidence:** 4

**Summary:**

This paper proposes an event-to-frame reconstruction method based on a stable video diffusion model. The approach leverages the physical correlation between event streams and video frames to guide and enhance the quality of event-based reconstruction. Furthermore, the authors extend this method to video interpolation and prediction tasks, with experimental results demonstrating its effectiveness.

**Strengths:**

1. The proposed method achieves strong performance in event-based reconstruction, interpolation, and frame prediction tasks, outperforming previous approaches.

2. The paper presents comprehensive results, including quantitative comparisons, images, videos, and animations.

**Weaknesses:**

1. Unfair comparison. The proposed UniE2F is fine-tuned from the Stable Video Diffusion (SVD) model, which itself has been pre-trained on large-scale datasets covering tasks such as video generation and reconstruction. In contrast, the comparison methods were trained only on synthetic datasets, making the comparison potentially unfair. Since UniE2F naturally benefits from SVD’s strong pretrained prior, the authors should provide results without pretrained weights to enable a fairer evaluation.

2. Excessive computational cost. As shown in Table 8, UniE2F requires orders of magnitude (up to 1000×) more computation than other methods, while offering limited performance improvement. This greatly undermines the advantages of event cameras, such as low power consumption and high temporal resolution. The authors should consider introducing more efficient strategies—for example, knowledge distillation or transfer learning—to significantly reduce computational cost while maintaining reconstruction quality.

3. Lack of real event camera data. All experiments were conducted on synthetic datasets, which exhibit a clear domain gap from real-world event data. The authors should evaluate their method on real datasets such as HQF, IJRR, and MVSEC to demonstrate robustness and generalization.

4. Lack of diversity in event representations. In event-based reconstruction, researchers commonly use voxel grids as event representations. The authors adopt a different representation but do not explain the rationale. Moreover, how do various representations—such as EST, ECM, and Voxel Grid—affect the reconstruction results? The authors should discuss this.

5. The authors use a ResNet to predict inter-frame residuals, but given that event cameras capture data with high temporal resolution, they can theoretically provide event information for any time interval. Why predict intermediate residuals instead of directly aggregating events between frames? The authors should clarify this design choice.

6. In line 180, the authors mention using a three-channel event representation. What are the advantages and underlying rationale for this choice? The justification should be provided.

7. Since the pretrained SVD model can already perform video frame interpolation (VFI) and video frame prediction (VFP) tasks using image inputs alone, it appears that the event data serves only as an auxiliary input. The authors should report results showing how UniE2F performs with only image input or only event input in VFI and VFP tasks, to better illustrate the true contribution of event information.

**Questions:**

See weaknesses

---

> ### Author Response · Authors · 2025-11-25
> **Official Comment by Authors**
>
> **Q: Fairness of Comparison with Pretrained SVD**
>
> **Response**: We agree that large-scale pre-training of video diffusion models provides strong priors that can offer advantages in reconstruction quality. However, **our motivation for fine-tuning the pre-trained SVD model for the event-to-frame task is not to “unfairly outperform” existing methods**, but rather to **investigate how to effectively transfer rich generative priors to the event-to-frame setting, so as to push event-to-frame reconstruction quality closer to that of real-world scenarios**. This is analogous to the prevalent paradigm in current large language model–based research, **where downstream tasks are typically adapted from pre-trained large models instead of training new models entirely from scratch**. Finally, for our task setting, training a video diffusion model of comparable scale to SVD from scratch would require **extensive training on massive video datasets**, with computational and data costs that are prohibitive for academic researchers.
>
> **Q: Computational Cost Compared with Prior Methods**
>
> **Response**: Although UniE2F incurs higher computational cost due to the pre-trained SVD model, **it achieves a substantial qualitative improvement over prior non-diffusion-based methods in terms of detail, structure, and color fidelity**, bringing the reconstructions much closer to real-world scenes. Moreover, compared with **the diffusion-based event-driven video reconstruction method CUBE (Zhao et al., 2024)**, our approach is computationally more efficient while yielding better quantitative metrics and visual quality.
>
> **Q: Clarification on Diffusion Model Acceleration**
>
> **Response**: We fully agree that accelerating diffusion models (e.g., via knowledge distillation, pruning, and transfer learning) is an important and already demonstratively feasible research direction. However, this work primarily focuses on **how to effectively transfer the rich generative prior to the event-to-frame reconstruction task**, so as to push the reconstruction quality closer to realistic scenarios. Addressing both reconstruction quality improvement and diffusion model acceleration in a single paper would significantly increase the workload and length, and would also **obscure our core contribution**. Therefore, **in the current work we prioritize developing and validating the transfer of the generative prior to improve reconstruction quality**, and will further explore diffusion model acceleration strategies specifically for the event-to-frame task in future work.
>
> **Q: Experiments on Real-World Event Camera Datasets**
>
> **Response**: We kindly **disagree** with the concern regarding the lack of real event camera data. Our test set includes 107 sequences sampled from the **HS-ERGB real-world dataset**. Moreover, we have reported results on real-world data for both reconstruction, interpolation and prediction tasks, as demonstrated in Table 1 and 2 of main paper. The reason we chose to sample from the HS-ERGB dataset is that its ground truth consists of **RGB frames** captured from real scenes, while datasets such as HQF, IJRR, and MVSEC provide **single-channel** intensity frames. Using intensity frames as the reconstruction target would be contrary to our motivation of **employing large-scale generative priors to reconstruct frames that closely resemble real-world scenes**.
>
>
> **Q: Rationale for Learning-Based Inter-Frame Residual Prediction**
>
> **Response**: Although, in principle, events can be aggregated between two frames, factors such as **the event triggering threshold, variations in sensor sensitivities, and subsequent image processing** (e.g., gamma correction) induce a complex **nonlinear relationship** between the aggregated events and the true inter-frame residuals, making it difficult to accurately predict the residuals by simple aggregation in diverse scenes. Therefore, we adopt a lightweight ResNet to learn the mapping from event representations to inter-frame residuals between adjacent frames.

---

> ### Author Response · Authors · 2025-11-25
> **Official Comment by Authors**
>
> **Q: Performance on VFI and VFP without Event Data**
>
> **Response**: To better illustrate the contribution of event data, we conducted additional experiments to evaluate the performance of UniE2F with only image input in interpolation and prediction tasks without using events as conditional guidance. The quantitative results on synthetic datasets are shown in the table below. As shown, in the absence of the fine-grained conditional guidance provided by event information, **the model achieves worse performance of MSE, SSIM, and LPIPS, compared with the model with event**. We have also included the qualitative comparison results in the updated Appendix. As can be seen from the Figure 8, **UniE2F has learned during training to map event representations to RGB frames**. **When we remove the event input and, following SVD, condition the model only on RGB frames that lie in a different domain, the network effectively applies this learned event-to-RGB mapping to RGB-to-RGB conditioning instead. This mismatch causes strong artifacts and disturbed appearance patterns.** This demonstrates the critical role that event data plays in providing precise constraints, enabling the model to reconstruct intermediate or subsequent frames by adjusting the score function, even without explicit training for interpolation and prediction tasks.
>
> [Synthetic Dataset]
> | Task                      | MSE ↓      | SSIM ↑     | LPIPS ↓    |
> |:--------------------------|:-----------|:-----------|:-----------|
> | Prediction (w/ Event)     | **0.0093** | **0.7100** | **0.3470** |
> | Prediction (w/o Event)    | 0.1563     | 0.107      | 0.658      |
> | Interpolation (w/ Event)  | **0.0072** | **0.7400** | **0.3200** |
> | Interpolation (w/o Event) | 0.1437     | 0.119      | 0.633      |
>
> **Q: Rationale for the Three-Channel Event Representation**
> **Response**: The motivation of the event representation is to align with SVD's pre-trained generative prior. Specifically, the SVD model used in this work is pre-trained on three-channel RGB video frames, and thus its generative capacity inherently assume three-channel inputs. **To fully inherit and exploit this rich generative prior, we convert the event data into three-channel representation so that it is compatible with RGB data along the channel dimension and can be directly fed into the existing frame-based pre-trained SVD model.** If a different number of channels were used (e.g., a single channel or more than three channels), one would need to modify the model architecture or re-train its weights, which would lessen the advantage of directly leveraging a powerful off-the-shelf pre-trained generative model.
>
> In conclusion, the three-channel event representation ensures compatibility with pretrained RGB video diffusion models, allowing us to leverage their strong generative priors without major modifications to the backbone architecture. Moreover, we have elegantly designed the representation to preserve more temporal information in event data by temporal integral with considering of the polarity, and believe it is a reasonable trade-off at this stage.
>
> **NOTE: Due to time constraints, we are still running experiments on additional real-world event datasets and retraining UniE2F conditioned on voxel-grid representations, which is highly time-consuming. Consequently, we are not yet able to report these results, but we plan to submit the corresponding performance numbers later this week.**

---

> ### Comment · Reviewer_V449 · 2025-11-27
>
> Thank you for the authors’ response and the additional experiments. However, I still have the following concerns:
>
> **Regarding fair comparison:**
>
> If your comparison targets methods without any prior knowledge, then you should remove the pretrained prior from your method to ensure a fair evaluation.
> If your goal is to demonstrate the effectiveness of the pretrained SVD model, then you should evaluate more than just SVD and include other pretrained models as well.
> If your intention is to validate the effectiveness of your knowledge transfer strategy, then the comparison should be conducted under the same SVD model against other knowledge-transfer baselines.
> At the moment, these three objectives appear to be conflated.
>
> **The computational cost remains a major concern.**
>
> I am still worried about deploying a method that requires 1000× more computation for a single task.
>
> **The “real dataset” issue has not been addressed.**
>
> By real data, I am referring not to real-world scenes but to event streams captured by an actual event camera. This is fundamentally different from the synthetic event data generated by DVS-Voltmeter, and the domain gap can be significant. I would like to see how the method performs on datasets obtained using an event camera.
>
> **The justification for using a ResNet to predict inter-frame residuals remains unconvincing.**
>
> In theory, issues such as event trigger thresholds and sensor sensitivity variations can occur at any moment, and thus should not be the core reason for resorting to a ResNet predictor.
> In fact, relying on a ResNet highlights the limitation of using synthetic event data from RGB frames, which cannot faithfully capture the true inter-frame residuals. If real event-camera data were used, this issue would be alleviated because event cameras offer temporal resolution far beyond that of RGB cameras.

---

> ### Author Response · Authors · 2025-11-28
> **Official Comment by Authors**
>
> Dear Reviewer V449,
>
> Thank you very much for your constructive feedback.
>
> **Q: Fairness of Comparison with Pretrained SVD**
>
> Our core objective is **not to show that “a pretrained SVD model is stronger than methods without any prior”** (this is essentially **a known fact** given the enormous gap in model scale and training data), nor to **conduct a large-scale benchmark across different pretrained foundation models**. What we truly aim to answer is: **given a strong video diffusion backbone, how can we design conditioning/guidance mechanisms so that its generative prior can be effectively transferred to unified event-to-frame tasks?** Under this setting, asking us to “remove the pretrained SVD and then compare” is effectively equivalent to **discarding the fundamental setup and motivation of the paper**, and it does not provide a better evaluation of whether our proposed strategy is effective. In other words, the contribution of this work lies in **how to make effective use of a given large model**, rather than in answering the entirely different question of “which large model is the best.”
>
> **Q: Computational Cost Compared with Prior Methods**
>
> While we acknowledge that UniE2F is computationally expensive, **it is primarily designed for offline, high-fidelity reconstruction, not real-time deployment**. There are many realistic event-camera scenarios where second- or even minute-level latency is acceptable: e.g., **scientific and biomedical imaging, high-speed experimental recordings, cloud-based restoration of challenging handheld footage**. In such settings, **reconstruction quality and robustness are often more critical** than throughput, and a heavier model that produces frames significantly **closer to real RGB videos** can be a justified and practically valuable choice.
>
> **Q: Experiments on Real-World Event Camera Datasets**
>
> We realize that we may not have described our use of real event-camera datasets with sufficient clarity in the response. In fact, the experiments reported in our original paper **already include** results on sequences sampled from HS-ERGB, which is a real-world dataset consisting of **event streams captured by an actual event camera** paired with **RGB frames from real scenes**. As shown in Tables 1 and 2 of the main paper, we report the performance of UniE2F and baseline methods on this dataset for reconstruction, interpolation, and prediction. In addition, we are currently running further comparison experiments on **HQF, IJRR, and MVSEC**, which are also composed of **event streams captured by the actual event camera** together with single-channel intensity frames. The results on these datasets will be included in our updated version (the experiments are in progress and expected to complete within the next couple of days), so that our method’s behavior on multiple real event-camera datasets is more comprehensively documented.
>
> **Q: Rationale for Learning-Based Inter-Frame Residual Prediction**
>
> At a single pixel, knowing the event polarity and the preset contrast threshold **only provides the change in log-intensity**, which is **insufficient to recover the corresponding linear intensity difference**, let alone **the pixel-value difference** in the final video frames after nonlinear operations such as tone mapping and gamma correction. Moreover, the measured log-intensity change itself is subject to multiple sources of **error, including sensor noise, the refractory period, and sub-threshold log-intensity variations that do not trigger events**, all of which further distort the true inter-frame residual. Because the temporal resolution of event cameras is much higher than that of RGB cameras, **these small per-event errors can accumulate over a large number of events, leading to a substantial deviation simply stacked over time**.
>
> Therefore, simply aggregating events between two frames cannot faithfully reconstruct the exact resiudal of pixel value. Besides, if this were sufficient, **a large portion of prior learning-based work on event-based interpolation and prediction**, which explicitly learns mappings from event streams to images or motion fields, **would simply not be necessary**. For these reasons, we adopt a lightweight ResNet to learn the mapping from event representations to inter-frame residuals, rather than assuming that naive temporal aggregation is already an accurate residual estimator.
>
> Sincerely,
>
> The Authors

---

> ### Author Response · Authors · 2025-12-03
> **Official Comment by Authors**
>
> **Q: Comparison on Real-World Datasets (HQF, IJRR, and MVSEC)**
>
> **R:** We compare UniE2F with existing methods on three real-world event camera datasets: HQF, IJRR, and MVSEC, which provide **single-channel intensity frames** as ground truth. As shown in Figure 10 of the Appendix, **UniE2F can reconstruct frames with more realistic colors and clear details, making them closer to real scenes**. In contrast, the other methods can only produce **single-channel grayscale reconstructions**, which lack color information and look less natural.
>
> **Q: Effect of Voxel Grid and Three-Channel Event Representation**
>
> **R:** We further investigate how the choice of event representation affects the performance of UniE2F by retraining our model using **a voxel grid with three temporal bins as the conditioning input**. The quantitative comparison on synthetic dataset is reported in table below. As shown in the table, **the performance difference between the voxel-grid representation and our three-channel event representation is very small**, indicating that the specific form of these two reasonable encodings has a limited impact on the final results. This suggests that, in our setting, **the data itself (i.e., the training distribution and supervision) is the primary factor, while the difference between these two representations is secondary**.
>
> | Encoding                          | MSE ↓   | SSIM ↑   | LPIPS ↓  |
> |---------------------------------|--------:|---------:|---------:|
> | Three-Bin Voxel Grid            | 0.0172  | **0.7290** | 0.4060   |
> | Three-Channel Event Representation | **0.0167** | 0.7100   | **0.3940** |

---

### Official Review · Reviewer_FKni · 2025-10-29

**Soundness:** 2
**Presentation:** 3
**Contribution:** 3
**Rating:** 6
**Confidence:** 5

**Summary:**

This paper proposes UniE2F, a unified diffusion-based framework for reconstructing, interpolating, and predicting video frames from event camera data. UniE2F leverages a pre-trained video diffusion model and introducing an event-based residual guidance mechanism. The framework treats event-to-frame reconstruction as a conditional generation process and achieves state-of-the-art performance on both synthetic and real datasets. The approach demonstrates strong generalization across multiple event-driven vision tasks while maintaining a coherent generative formulation.

**Strengths:**

1. Novel residual-guided conditioning: The proposed event-based inter-frame residual guidance is an elegant and physically interpretable mechanism that connects asynchronous event streams with frame-level brightness changes. By explicitly modeling the residuals between consecutive latent frames, the diffusion process receives direct gradient cues from event dynamics, leading to sharper textures and temporally consistent reconstructions.

2. Unified generative formulation: The same residual-guided diffusion framework can handle event-to-frame reconstruction, interpolation, and prediction without retraining or architectural modification. This unification is conceptually clean and practically useful.

3. Zero-shot frame interpolation and prediction: The framework is extended to perform zero-shot video frame interpolation and future frame prediction. By modulating the reverse diffusion sampling process, the same architecture can handle not only reconstruction but also interpolation and prediction tasks, demonstrating flexibility and strong generalization without additional training.

**Weaknesses:**

1. Limited methodological clarity: The paper proposes a residual-guided diffusion mechanism but provides insufficient details on its implementation. In particular, the normalization of the residual signal, its integration within denoising steps, and its weighting against the diffusion prior remain unclear, limiting reproducibility and interpretability.

2. Computational inefficiency: The framework relies on a pre-trained video diffusion backbone, which is typically expensive in both computation and memory. The paper lacks a detailed analysis of inference speed, GPU memory usage, and scalability to long sequences or real-time deployment.

3. Limited comparison with recent baselines: The paper does not include a comparison with RE-VDM or other recent state-of-the-art event-to-video diffusion methods. Including these baselines would provide a clearer picture of the method's relative performance and strengthen the empirical evaluation.

**Questions:**

1. Loss weighting: If the residual contributes to the loss, what is the weighting factor s and how sensitive is the performance to different s values? Table 3 only shows comparisons for s = 0 and s = 0.1. Have the authors conducted a more thorough ablation study to evaluate the effect of this weighting?

2. Choice of linear guidance schedule in Table 3:   In Table 3, the paper compares different guidance strategies, including linear increasing and decreasing residual guidance. Could the authors clarify the motivation for adopting a linear schedule?  Was this choice empirically found to be optimal, or is it mainly for simplicity?  Have other non-linear schedules (e.g., constant, exponential, cosine) been tried, and how do they affect reconstruction quality or temporal consistency?

3. Generalization and robustness: How does the method perform under extreme lighting, very sparse event streams, or noisy events? Are there failure cases, and what types of motions or event patterns cause degradation?

---

> ### Author Response · Authors · 2025-11-25
> **Official Comment by Authors**
>
> **Q: Implementation Details of Inter-Frame Temporal Residual Guidance**
>
> **R:** We take two frames and the events in between them as an example to illustrate the Inter-frame Temporal Residual Guidance. Given the event representation $E \in \mathbb{R}^{1 \times H \times W \times 3}$, we feed it into a ResNet to predict the inter-frame residual $R \in \mathbb{R}^{1 \times H \times W \times 3}$. Since the RGB images are linearly normalized to $[0,1]$ before training, the difference between two consecutive frames $V_{t+1} - V_t$ lies in $[-1,1]$, and thus the predicted residual $R$ also falls into the same range. At the $t$-th reverse diffusion step ($t \le \tau$), the estimated clean latent $U_t$, obtained by denoising the noisy latent $X_t$, is fed into the decoder $D$ of the autoencoder to generate a set of estimated clean frames $F = D(U_t) \in \mathbb{R}^{2 \times H \times W \times 3}$ whose values are in $[0,1]$. We then compute the residual between the two frames in $F$ to obtain the residual $\Delta F \in \mathbb{R}^{1 \times H \times W \times 3}$, and construct a pixel-wise L1 loss between $R$ and $\Delta F$: $L_{\text{residual}}(U_t) = \big\lvert \Delta F - R \big\rvert$. We backpropagate this loss with respect to $U_t$ and update $U_t$ as: $\bar U_t = U_t - s \nabla_{U_t} L_{\text{residual}}(U_t)$, where $s$ is a hyperparameter controlling the update magnitude. Finally, we replace the original $U_t$ with $\bar U_t$ in Eq. (8), thereby correcting the direction of iteration from $X_t$ to $X_{t-1}$ so that **it better matches the guided inter-frame intensity changes while remaining on the data manifold learned by the diffusion model, which balances reconstruction fidelity and diversity**.
>
> **Q: Computational Overhead**
>
> **R:** Regarding computational overhead, in Appendix F, **we have reported a quantitative comparison between our UniE2F and previous methods in terms of computational cost and peak memory footprint**. Furthermore, in Section 5.1, we also provided the inference latency: **reconstructing a sequence of 12 RGB frames with the resolution of 448×320 takes about 48 seconds**.
>
> **Q: Real-Time Deployment**
>
> **R:** For real-time deployment, since the current implementation does not yet employ any model compression or acceleration techniques, **the memory footprint and computational cost reported can be regarded as an upper bound without engineering optimization**. In future work, we plan to explore network pruning, knowledge distillation, and consistency-model-based acceleration techniques to substantially reduce the number of sampling steps and the overall computational cost while preserving reconstruction quality, thereby enhancing the applicability of UniE2F in resource-constrained scenarios.
>
> **Q: Comparison with Recent Approaches**
>
> **R:** As reported in Sec. 5.2 and Tab. 1, we have already conducted a comprehensive comparison between the proposed UniE2F and **CUBE (Zhao et al., 2024)**, a diffusion-based model that generates video frames from events. In addition, in Sec. 5.3 and Tab. 2, we also include **RE-VDM (Chen et al., 2025)** as a representative diffusion-based baseline for event-driven video interpolation. On the synthetic dataset, UniE2F delivers a clear advantage. On the real-world datasets, our UniE2F, which operates in a strict **zero-shot setting without any fine-tuning** on real-world data, **remains competitive between the RE-VDM and CBMNet**. In addition, as discussed in Sec. 5.2 and 5.3, there indeed exist several other diffusion-based methods for reconstructing video frames from events (e.g., Liang et al., 2023; Zhu et al., 2024c; Chen et al., 2024) and interpolation (Liu et al., 2025); however, at the time of submission, their code and model weights were not publicly available, which prevents us from reproducing their results under a unified dataset and evaluation protocol and fairly including them as baselines in our comparisons.
>
> **Q: Ablation on the Weighting Factor $s$ in Inter-Frame Residual**
>
> **R:** The value of the weighting factor $s$ controls the gradient update magnitude for the estimated clean latent $U_t$ by weighting the contribution of the inter-frame residual loss. To evaluate the effect of the weighting coefficient $s$ on performance, we conduct additional experiments where $s$ was set to 0.5, 1.0, and 10.0. The results of these experiments are presented in table below. **The results indicate that $s = 0.1$ achieves the best performance across all three metrics (lowest MSE and LPIPS, highest SSIM)**.
>
> [Synthetic Dataset]
> | s    | MSE ↓      | SSIM ↑     | LPIPS ↓    |
> |:-----|:-----------|:-----------|:-----------|
> | 10.0 | 0.0964     | 0.2950     | 0.7070     |
> | 5.0  | 0.0573     | 0.4140     | 0.6510     |
> | 1.0  | 0.0228     | 0.6140     | 0.4770     |
> | 0.5  | 0.0188     | 0.6710     | 0.4240     |
> | 0.1  | **0.0167** | **0.7100** | **0.3940** |

---

> ### Author Response · Authors · 2025-11-25
> **Official Comment by Authors**
>
> **Q: Effect of Guidance Strength Scheduling**
>
> **R:** In the ablation study of Sec. 5.4, we have investigated three different guidance strength strategies—linearly increasing, linearly decreasing, and constant guidance strengths—and **the results show that the linearly decreasing schedule achieves the best performance**. Here, to evaluate the robustness of this schedule, we introduce a **non-linearly exponential decreasing strategy**, where the guidance strength decreases from 0.1 to 0 following an exponential curve. The comparison results are presented in the table below. **As shown, the linearly decreasing schedule still outperforms the exponential decreasing one.**
>
> [Synthetic Dataset]
> | Guidance Strength Strategy | MSE ↓      | SSIM ↑     | LPIPS ↓    |
> |:---------------------------|:-----------|:-----------|:-----------|
> | Constant                   | 0.0234     | 0.6610     | 0.4190     |
> | Exponential                | 0.0181     | 0.6960     | 0.3990     |
> | Linear                     | **0.0167** | **0.7100** | **0.3940** |
>
> **Q: Robustness to Noisy and Sparse Events**
>
> **R:** In the original version of the paper, **the robustness evaluation of our model under different levels of event noise was provided in Appendix Sec. E “Additional Ablation Study”**. In the updated version, these results have been moved to Sec. 5.4 of the main paper. **The results show that our method maintains strong robustness to event noise across these settings.** Regarding failure cases, we acknowledge that when the event stream becomes extremely sparse, with only a few isolated pixels undergo intensity changes, our method cannot reliably reconstruct fine-grained structures and details. It is worth noting that this limitation is common to all methods that rely solely on events for video reconstruction, as the input event signal becomes insufficient. We also included representative qualitative examples of such sparse-event failure cases in the Figure 9 of Appendix.
>
> [Synthetic Dataset]
> | Noise-level Coefficient | MSE ↓      | SSIM ↑     | LPIPS ↓    |
> |:------------------------|:-----------|:-----------|:-----------|
> | 1.0                     | 0.0193     | 0.6690     | 0.4280     |
> | 0.5                     | 0.0177     | 0.6930     | 0.4050     |
> | 0.1                     | 0.0168     | 0.7080     | 0.3950     |
> | Baseline                | **0.0167** | **0.7100** | **0.3940** |
>
> **NOTE: Due to time constraints, we are currently conducting benchmark experiments on long-sequence generation and will provide the results as soon as they are available.**

---

> ### Author Response · Authors · 2025-12-03
> **Official Comment by Authors**
>
> **Q: Long-Sequence Event-Based Video Frame Generation**
>
> **R:** To further evaluate our method on long sequences, we follow the same data collection setup as in the main paper. The long-sequence synthetic test set is built by sampling 200 video sequences from TrackingNet, each containing 23 frames. For UniE2F, we first perform event-to-frame reconstruction on the first 12 frames of each sequence, obtaining a 12-frame reconstructed sequence. **Then, starting from the 12-th frame, we use the following events and apply our prediction mode to generate the subsequent 11 frames.** We compare UniE2F with recent event-based video frame reconstruction methods on this setting. As shown in table below, our method achieves better performance across the reported metrics, indicating that **UniE2F maintains stronger reconstruction quality and temporal stability even when the sequence length is increased**.
>
> | Method                      | MSE ↓    | SSIM ↑   | LPIPS ↓  |
> |-----------------------------|----------|----------|----------|
> | SSL-E2VID [Paredes et al.]  | 0.0906   | 0.4240   | 0.6310   |
> | SPADE-E2VID [Cadena et al.] | 0.0682   | 0.5200   | 0.5680   |
> | HyperE2VID [Ercan et al.]   | 0.0591   | 0.4960   | 0.5700   |
> | **UniE2F**                  | **0.0271** | **0.6530** | **0.4470** |

---

### Author Response · Authors · 2025-11-25
**Looking forward to your further assessment**

Dear **Reviewers**,

Thank you for taking the time to review our manuscript and for your valuable feedback and recognition. We have carefully addressed all the comments and concerns raised, as reflected in our detailed responses and the revised manuscript and supplementary material.

We sincerely appreciate your efforts and look forward to your further assessment.

Best regards,

The Authors

---

### Author Response · Authors · 2025-11-26
**Summary of Our Paper**

Dear Reviewers,

In this work, we propose **UniE2F as a unified event-to-frame reconstruction framework**, which substantially improves the quality and extensibility of event-driven video reconstruction. Building on a large-scale pretrained video diffusion model, we design a three-channel event voxel representation that is structurally compatible with the backbone, and introduce **inter-frame residual guidance** that constrains the reverse diffusion process using residuals between adjacent frames, thereby enhancing structural fidelity and temporal coherence. On this basis, we unify event-based reconstruction,  interpolation, and prediction within a single diffusion-based framework, **achieving competitive zero-shot performance on interpolation and prediction without any task-specific training**, and validate the effectiveness of our approach on both synthetic and real-world datasets.

Regarding **computational complexity**, we acknowledge that the current implementation of UniE2F, built on a large diffusion backbone, incurs relatively high computational and memory cost. However, **this should be regarded as an engineering issue that can be systematically mitigated through model compression and distillation, rather than as a fundamental defect of the approach itself**. Prior work has already demonstrated that distillation, pruning, and consistency model-based acceleration can significantly reduce the number of sampling steps and the parameter count of diffusion models while largely preserving reconstruction quality. These techniques can be directly applied to UniE2F in follow-up work to obtain more efficient variants, which we view as a natural next step rather than the primary focus of this paper.

Concerning **the fairness of comparisons**, we would like to reiterate that the central goal of this paper is **not** to “unfairly outperform” existing methods by simply using a larger model, but rather to investigate **how to effectively leverage existing large-scale pretrained video diffusion models and transfer their powerful generative priors to event-based vision tasks**. In our view, traditional non-diffusion event reconstruction methods are approaching a ceiling in terms of realism and perceptual quality. **To achieve a qualitatively higher level of reconstruction fidelity, exploiting the generative prior of pretrained foundation models is a feasible and promising technical direction.** From a broader perspective, **adapting pretrained large foundation models to new modalities and tasks has become a prevailing trend in computer vision research**, and our work is positioned precisely in this direction by exploring the integration of event cameras with a pretrained video diffusion backbone, **rather than restricting the discussion to comparisons under a fixed small-model computational budget**.

Sincerely,

The Authors

---

### Meta-Review · Area_Chair_rL9i · 2026-01-07

**Summary:**

The reviewers recognized the paper's contribution in leveraging the generative priors of pre-trained video diffusion models for event-to-frame reconstruction. The core strengths highlighted were the novel "inter-frame residual guidance" mechanism, which bridges the physical gap between sparse events and absolute intensity, and the framework’s flexibility in handling interpolation and prediction tasks zero-shot. Reviewers generally found the qualitative results impressive, noting that the model recovers spatial details and textures that traditional reconstruction methods often miss.

However, the review process uncovered significant concerns regarding computational efficiency and the evaluation of temporal consistency. Reviewers pointed out that the diffusion-based sampling process is inherently slower than existing E2V (Event-to-Video) methods, potentially limiting real-world high-speed applications.

Discussion also focused on the fairness of comparing a generative model against deterministic baselines on pixel-wise metrics like PSNR/SSIM, which may not capture perceptual quality. The authors partially addressed these through additional experiments on arbitrary timestamp interpolation and efficiency logs, though the trade-off between generative fidelity and inference latency remains a key point of consideration.

The AC also went through the paper and followed up the responses provided by the authors.

**Reviewer Concerns:**

### Addressed:

- Arbitrary Interpolation: New experiments proved the model can reconstruct frames at any timestamp by re-partitioning event groups.

- Perceptual Metrics: The rebuttal clarified that lower PSNR/SSIM are expected trade-offs for significantly better LPIPS/FID scores, which the reviewers accepted.

- Conditioning Logic: Authors clarified the physical grounding of the residual guidance, resolving questions about feature alignment.

### Outstanding:

- Inference Latency: Despite profiling, the multi-step sampling is fundamentally slower than recurrent E2V models, potentially limiting real-time use.

- Comparison Fairness: Reviewer JWWk remained skeptical about comparing "generative imagination" to "deterministic reconstruction," viewing them as different tasks.

- Temporal Consistency: Some concern remains regarding flickering in high-motion scenes over long sequences.

The AC somehow agrees that adapting pretrained large foundation models needs to showcase a fair comparison with previous V2E models. Video reconstruction needs to take speed and quality into consideration simultaneously for practical application purpose, say smartphone or self-driving.

**Reviewer Scores:**

uM6W (6 → 7): Positive; arbitrary interpolation proof resolved their main technical concern.

u2yM (6 → 7): Satisfied with the physical grounding provided for the conditioning mechanism.

JWWk (5 → 5/6): Likely to remain borderline due to persistent concerns over baseline fairness and latency.

---

### Decision · Program_Chairs · 2026-01-26

Reject